# B cell heterogeneity in human tuberculosis highlights compartment-specific phenotype and functional roles
Robert Krause [1,2] ✉, Paul Ogongo [1,2,3], Liku Tezera[4,5,6], Mohammed Ahmed[1,2], Ian Mbano[1,2], Mark Chambers[1,2], Abigail Ngoepe[1], Magalli Magnoumba[1,2], Daniel Muema[1,2], Farina Karim[1,2], Khadija Khan [1,2], Kapongo Lumamba[1], Kievershen Nargan[1], Rajhmun Madansein[7], Adrie Steyn [1,2,8,9], Alex K. Shalek [10,11,12], Paul Elkington[4,5] & Al Leslie [1,2,6] ✉

B cells are important in tuberculosis (TB) immunity, but their role in the human lung is understudied. Here, we characterize B cells from lung tissue and matched blood of patients with TB and found they are decreased in the blood and increased in the lungs, consistent with recruitment to infected tissue, where they are located in granuloma associated lymphoid tissue. Flow cytometry and transcriptomics identify multiple B cell populations in the lung, including those associated with tissue resident memory, germinal centers, antibody secretion, proinflammatory atypical B cells, and regulatory B cells, some of which are expanded in TB disease. Additionally, TB lungs contain high levels of *Mtb*-reactive antibodies, specifically IgM, which promotes *Mtb* phagocytosis. Overall, these data reveal the presence of functionally diverse B cell subsets in the lungs of patients with TB and suggest several potential localized roles that may represent a target for interventions to promote immunity or mitigate immunopathology.

In 2021, *Mycobacterium tuberculosis* (*Mtb*) was estimated to infect a quarter of the world's population, cause tuberculosis (TB) in 10.6 million and kill 1.6 million people[1]. *Mtb* is spread when exhaled/coughed aerosolized droplets are inhaled into another individual's lungs where infection propagates. A characteristic of pulmonary TB is the generation of granuloma[2], typically formed from clusters of infected alveolar macrophages surrounded by a cuff of lymphocytes, including T cells, B cells and various innate lymphoid cells. In addition, B cells are frequently observed within lymphocyte aggregates near granuloma, referred to as granuloma-associated lymphoid tissue (GrALT)[3]. GrALT has been observed in mice[4–8], non-human primates (NHP)[8–10] and humans[8,11,12] and is typically associated with protective immunity. However, the role of B cells in the immune response to TB within the lung remains poorly understood.

In humans, TB leads to a reduction in circulating B cells[11,13–17], which recover following successful treatment[11,15,18]. In mice, B cell knock out or depletion results in greater susceptibility to TB[4,6], with adoptive transfer resulting in reversed lung pathology[7,19]. In non-human primates, B cell depletion increases bacterial burden in lung lesions[10]. Mechanistically, recent studies implicate B cells in orchestrating the CD4+ T cell response within GrALT, affecting both the $T_H1$ and $T_H17$ protective responses[3,20–23]. In addition, B cell follicle area correlated with lung *IL17* mRNA levels, and B cells from pleural fluid were critical mediators of the IL17 and IL22 responses[24–26]. Tissue-resident memory B cells reside in the lung and can mature into antibody-secreting plasma cells[27]. Several studies demonstrate a role for *Mtb*-specific antibodies in protective immunity to TB in both NHPs and humans[22,23,28–32]. Furthermore, active and latent TB can be differentiated

[1]Africa Health Research Institute, Durban, South Africa. [2]School of Laboratory Medicine and Medical Sciences, University of KwaZulu-Natal, Durban, South Africa. [3]Institute of Primate Research, National Museums of Kenya, Nairobi, Kenya. [4]National Institute for Health Research Southampton Biomedical Research Centre, School of Clinical and Experimental Sciences, Faculty of Medicine, University of Southampton, Southampton, UK. [5]Institute for Life Sciences, University of Southampton, Southampton, UK. [6]Division of Infection and Immunity, University College London, London, UK. [7]Department of Cardiothoracic Surgery, Nelson Mandela School of Medicine, University of KwaZulu-Natal, Durban, South Africa. [8]Department of Microbiology, University of Alabama at Birmingham, Birmingham, AL, USA. [9]Center for AIDS Research and Center for Free Radical Biology, University of Alabama at Birmingham, Birmingham, AL, USA. [10]Institute for Medical Engineering & Science, Department of Chemistry, Koch Institute for Integrative Cancer Research, Massachusetts Institute of Technology, Cambridge, MA, USA. [11]Ragon Institute of MGH, MIT and Harvard, Cambridge, MA, USA. [12]Broad Institute of MIT and Harvard, Cambridge, MA, USA. ✉e-mail: Robert.krause@ahri.org; al.leslie@ahri.org

based on antibody glycosylation profiles[29,33] and the resulting differences in the Fc effector functions (binding FcγRIII) including antibody-mediated phagocytosis[29].

Together, these studies demonstrate B cell involvement during TB and highlight important canonical and non-canonical functions that contribute to TB immunity. However, data from humans in the lung compartment is lacking. Here, we demonstrate that TB-infected lung tissues are enriched for CD19[+] B cells, which are associated with granuloma. Lung B cells were predominantly of a memory phenotype and expressed markers associated with residency and germinal centre homing. In addition, the lung was enriched for antibody-secreting cells (ASC) and unique putative regulatory B cell phenotypes. Using a granuloma biomimetic model, we found that B cells from healthy donors contribute to control of *Mtb* growth, while this function is inconsistent in patients with TB, suggesting potential impairment of these cells during active disease. Finally, the lungs of patients with TB were enriched for *Mtb*-specific antibodies relative to control lungs,

especially IgM, which enhanced *Mtb* phagocytosis by primary human cells. These findings highlight the diversity of B cell function in human TB.

## Results

### Comparison of CD19[+] B cells in blood and lung tissue compartments

First, we compared the relative frequencies of B cells in the peripheral blood of non-TB control patients, patients with latent TB (LTBI) and active TB (ATBI) (Fig. 1a, b). LTBI was confirmed by a positive Interferon Gamma Release Assay (IGRA) test in participants with no signs, symptoms, or history of TB disease. Non-TB controls were similarly asymptomatic but were IGRA negative[16]. The frequency of CD19[+] B cells was reduced in the blood of LTBI and ATBI compared to healthy controls, consistent with published data[11,13–16]. We then measured the frequency of B cells in homogenized lung tissue from patients with TB, obtained from individuals enrolled in the AHRI lung resection cohort[34], undergoing surgical resection

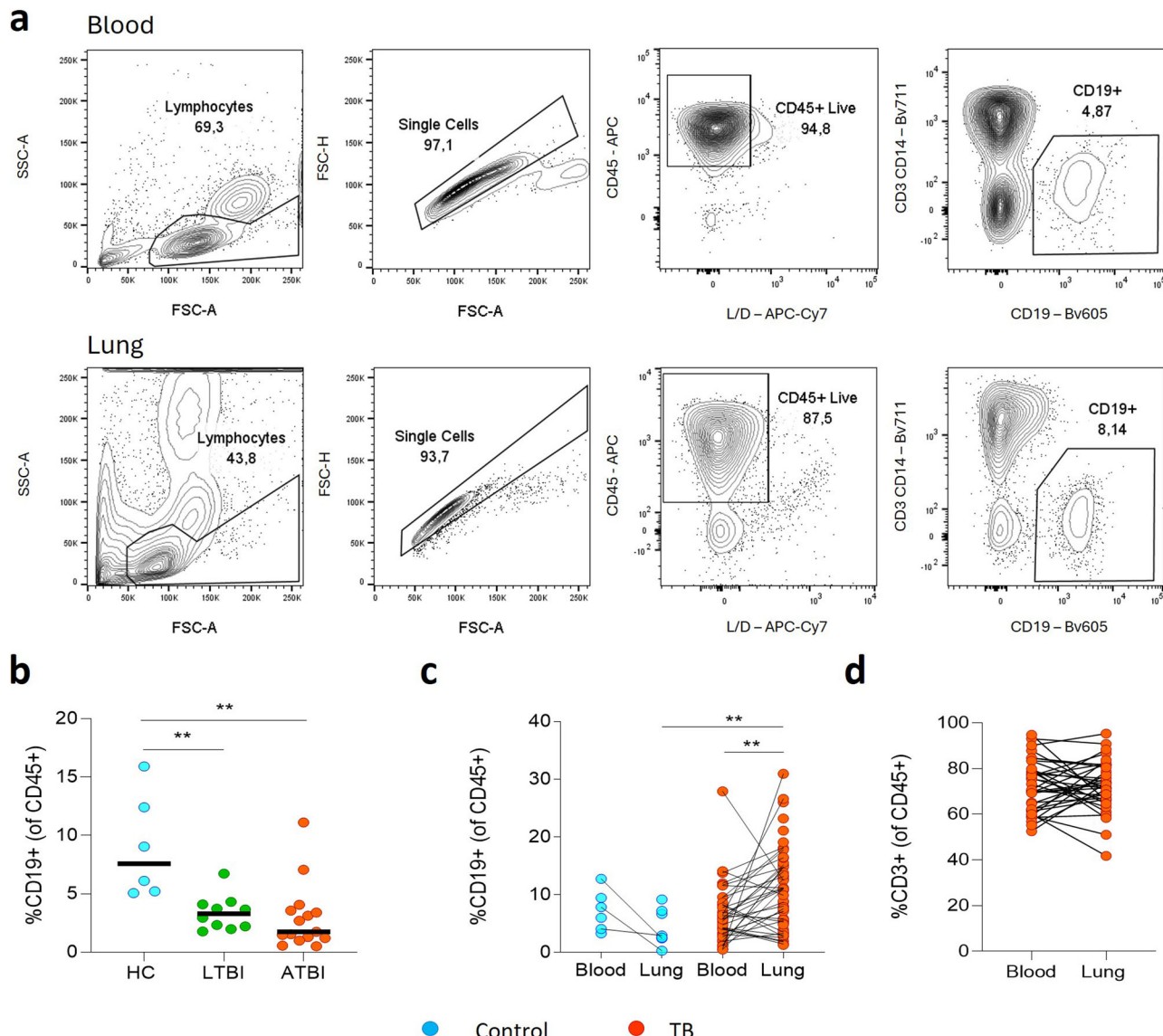

**Fig. 1 | Frequencies of CD19[+] B cells differ between blood and lung compartments. a** Representative flow plots identifying CD19[+] B cells from peripheral blood mononuclear cells (PBMC) and resected lung tissue. **b** B cell frequencies in the blood were compared for a set of healthy controls (HC, $n = 6$), patients with latent TB (LTBI, $n = 10$) or active TB (ATBI, $n = 15$), demonstrating reduced circulating B cells in active TB. **c** B cell frequencies were compared for lung tissue versus matched blood of cancer control patients (in cyan, $n = 8$) and patients with TB (in red, $n = 60$). B cell numbers were increased in the lung compared to the blood in patients with TB, and also relative to cancer control lung tissue. **d** In contrast, comparative frequencies of CD3[+] T cells between lung tissue and matched blood of patients with TB showed no significant difference. Statistical analyses were performed using the Mann–Whitney test between unmatched samples and the Wilcoxon test for matched/paired samples. P values are denoted by ** <0.01.

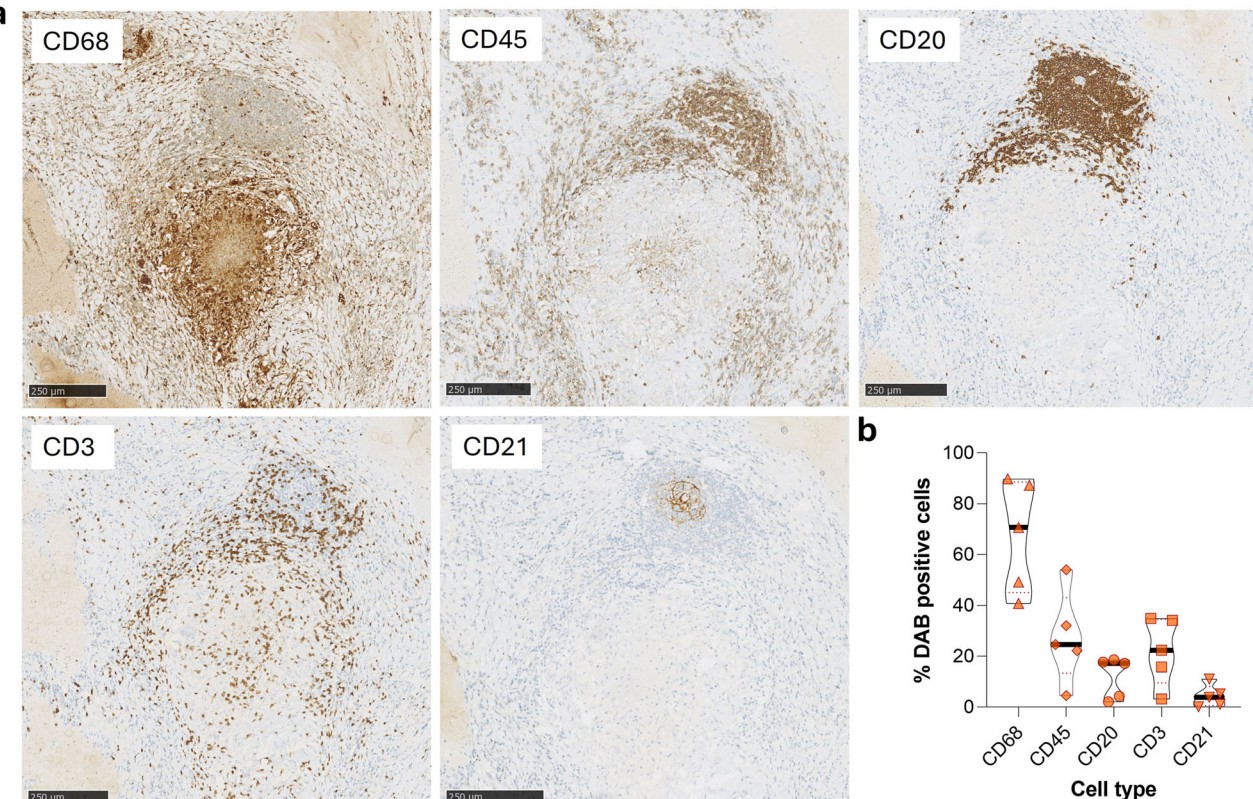

**Fig. 2 | Immunohistochemistry of human lung tissue resection from a patient with TB demonstrates B cell aggregates adjacent to the granuloma. a** Serial sections of lung tissue from patients with TB were stained for macrophages (anti-CD68), leucocytes (anti-CD45), B cells (anti-CD20), T cells (anti-CD3) and dendritic cells (anti-CD21). **b** The diaminobenzidine (DAB) stained cells were enumerated and expressed as % DAB positive cells of total nucleated (haematoxylin+) cells, per total slide area imaged. The relative frequencies from a total of five different tissue samples from patients with TB are compared.

to treat TB sequelae (Fig. 1a, c). Additional samples were analysed from patients with no history of TB, who were undergoing surgical resection of lung cancer and from whom macroscopically uninvolved tissue was obtained. These controls were not IGRA tested, and thus latent TB infection could not be excluded, although TB prevalence is so high in KwaZulu-Natal, most donors are highly likely to have been TB-exposed. The lung tissue of patients with TB was highly enriched for B cells compared to matched blood samples, with a median frequency of approximately 9% of CD45+, reaching as high as 31% in some patients (Fig. 1c), significantly higher than in lung tissue from non-TB controls (median of 3%). In contrast, CD3+ T cells were not significantly enriched in lung tissue from patients with TB compared to matched blood, although these cells were more abundant than B cells (median 72%, Fig. 1d).

**Histological localization of B cells in proximity to granulomatous structures**

To determine the localization of B cells within lung tissue of patients with TB, we conducted histological assessment of lung sections from participants with distinct granulomatous lesions. The canonical TB lung granuloma has a CD68+ macrophage core surrounded by a cuff of CD45+ lymphocytes (Fig. 2a). This cuff comprises CD3+ T cells with some also present within the granuloma core. CD20+ staining reveals a distinct B cell aggregate in close association with the granuloma, consistent with previously described GrALT[8,11,12]. B cells also associate with the granuloma cuff, primarily adjacent to the GrALT. The GrALT also contains CD3+ T cells, consistent with previous reports and with lymphoid follicles. In addition, CD21 staining is present within the centre of the GrALT, suggestive of follicular dendritic cells[8,35], although mature B cells can express CD21 within germinal centres[36]. Of interest were the staining patterns of anti-CD20, anti-CD3 and anti-CD21 in particular (Supplementary Fig. 1).

CD20 and CD3 staining are characteristic of lymphoid cells whereas CD21 stained cells displaying a more dendrite-like pattern. The CD20+ cells occupy most of the aggregate, whereas the CD3+ cells are more peripheral, and the CD21+ cells seem to occupy the centre of the aggregate. Quantitation of granulomatous lesion containing lungs from 5 different patients with TB show a roughly equivalent frequency of T and B cells associated with lung granuloma, in the region of 20% of nucleated (haematoxylin+) cells (Fig. 2b). As some surface markers are co-expressed by different cells, the total exceeds 100%. Overall, these data suggest that B cells are enriched in the lung during TB infection in humans and make up a significant proportion of granuloma-associated lymphocytes.

**Prominent memory and antibody-secreting cell (ASC) phenotypes in the lung compartment including CD69+ memory**

Next, we examined the phenotype of B cells in matched blood and lung samples from participants with TB and non-TB controls (Fig. 3). First, we examined the B cell maturation state based on the canonical memory marker CD27 and activation marker CD38[37] (Fig. 3a). Notably, CD27+CD38- memory B cells were enriched in lung tissue from patients with TB compared to matched blood, which was dominated by CD27-CD38- naïve and CD27-CD38+ transitional B cells (Fig. 3b). The same trend was observed in control lung samples, although the enrichment of memory B cells was not significant and memory B cells were significantly more abundant in lungs from patients with TB compared to controls. CD27+CD38hi antibody-secreting cells (ASCs) were present in lung tissue, and were significantly enriched compared to matched blood, but significantly higher in non-TB control lung tissue. In addition, lung derived memory B cells were highly enriched for the expression of the tissue residence marker CD69[27] (Fig. 3c). Finally, from a subset of lung samples, lung draining lymph nodes were studied as an alternative comparator tissue

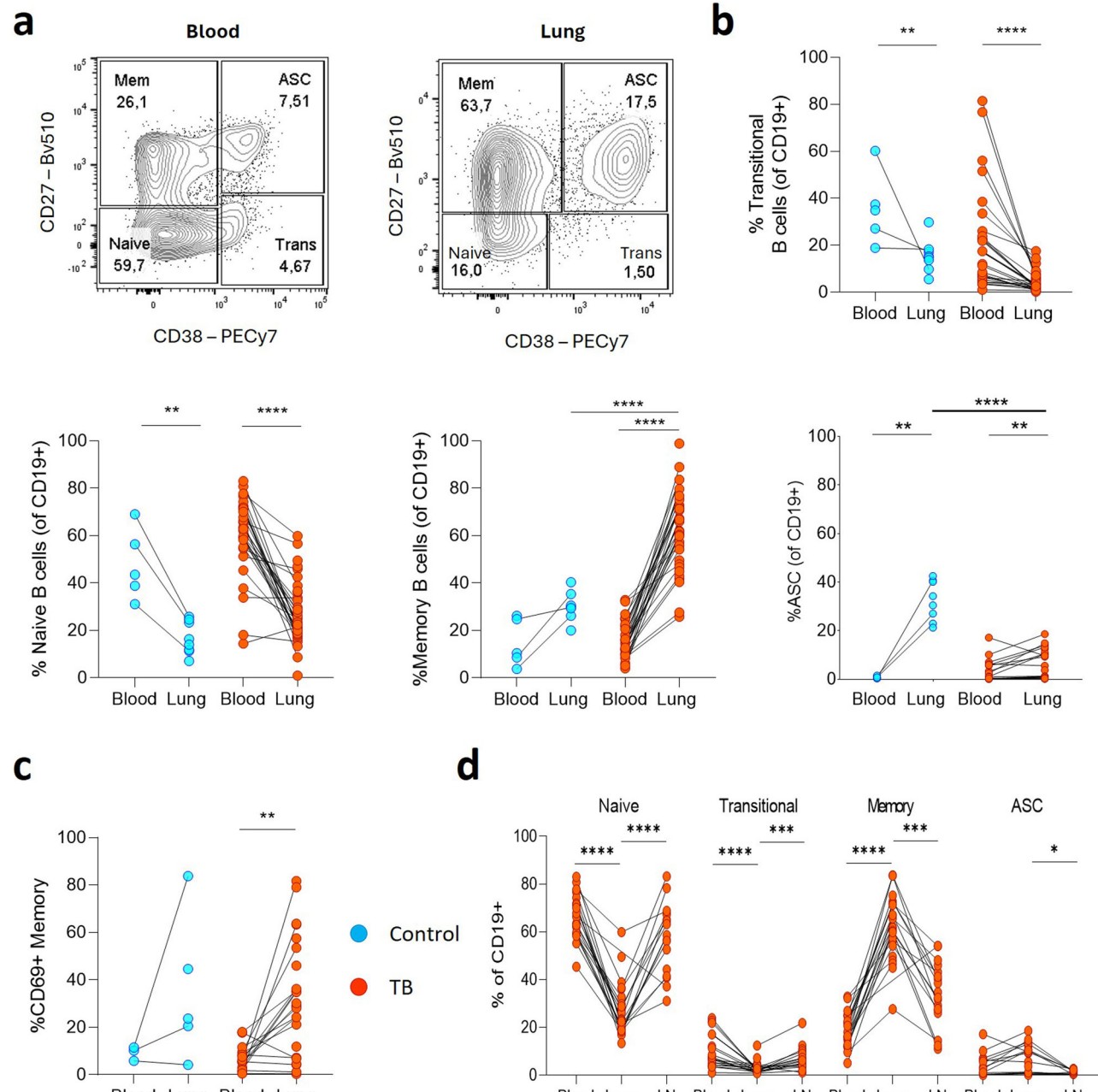

**Fig. 3 | B cell transitional, naive, memory and antibody-secreting cell population frequency demonstrates differences between compartments that are mainly disease-independent. a** Representative flow plots of the canonical transitional (Trans), naive, memory (Mem) and antibody-secreting cell (ASC) B cell phenotypes derived from staining CD19+ B cells with CD27 and CD38 (cancer control in cyan, n = 8; patients with TB in red, n = 60). **b** The relative frequencies of these four B cell phenotypes were compared between blood and lung tissue compartments. **c** The frequency of CD69+ memory B cells was assessed (cancer control in cyan, n = 5;

patients with TB in red, n = 25). **d** This analysis was extended to lung draining lymph nodes (LN, n = 17) from the TB cohort for comparison. A representative gating strategy for LN derived B cells is included as Supplementary Fig. 2. Statistical analyses were performed using the Mann–Whitney test between unmatched samples, the Wilcoxon test for matched/paired samples and a three-way Kruskal–Wallis comparison. P values are denoted by * ≤0.05; ** <0.01; *** <0.001 and **** <0.0001.

enriched with B cells (Fig. 3d, Supplementary Fig. 2)[38]. Lung tissue had increased memory subsets compared to matched lymph nodes, and as expected, lymph nodes contained a low prevalence of ASCs[38].

### Transcriptional analysis of lung-derived B cells

Having observed an enrichment of memory B cell subsets in lung tissue from patients with TB, we interrogated these in a single cell RNA sequence (scRNAseq) analysis of additional lung tissue samples obtained from the same lung cohort. Of a total 20962 single cells recovered from 9 resected lungs from patients with TB, 404 displayed canonical B cell markers including the

lineage markers CD19 and CD20. We focused on leucocytes, and sub clustering of this limited dataset identified a total of 7 unique B cell subsets, shown by UMAP projection (Fig. 4a). A curated list of B cell genes was used to assign putative functional phenotypes to the 7 subsets (Fig. 4b). This includes populations expressing genes associated with memory B cells (3 and 4) which expressed CD21, a B cell coreceptor involved in T cell-dependent signalling, in addition to CD27 and CD38, and ASCs (5 and 6), expressing the highest levels of genes associated with immunoglobulin heavy chains, including subclasses IgG 1-4 as well as IgA. These cells also expressed CD138 and BLIMP1, both markers of long-lived plasma cells[37,39], and IgJ and PPIB,

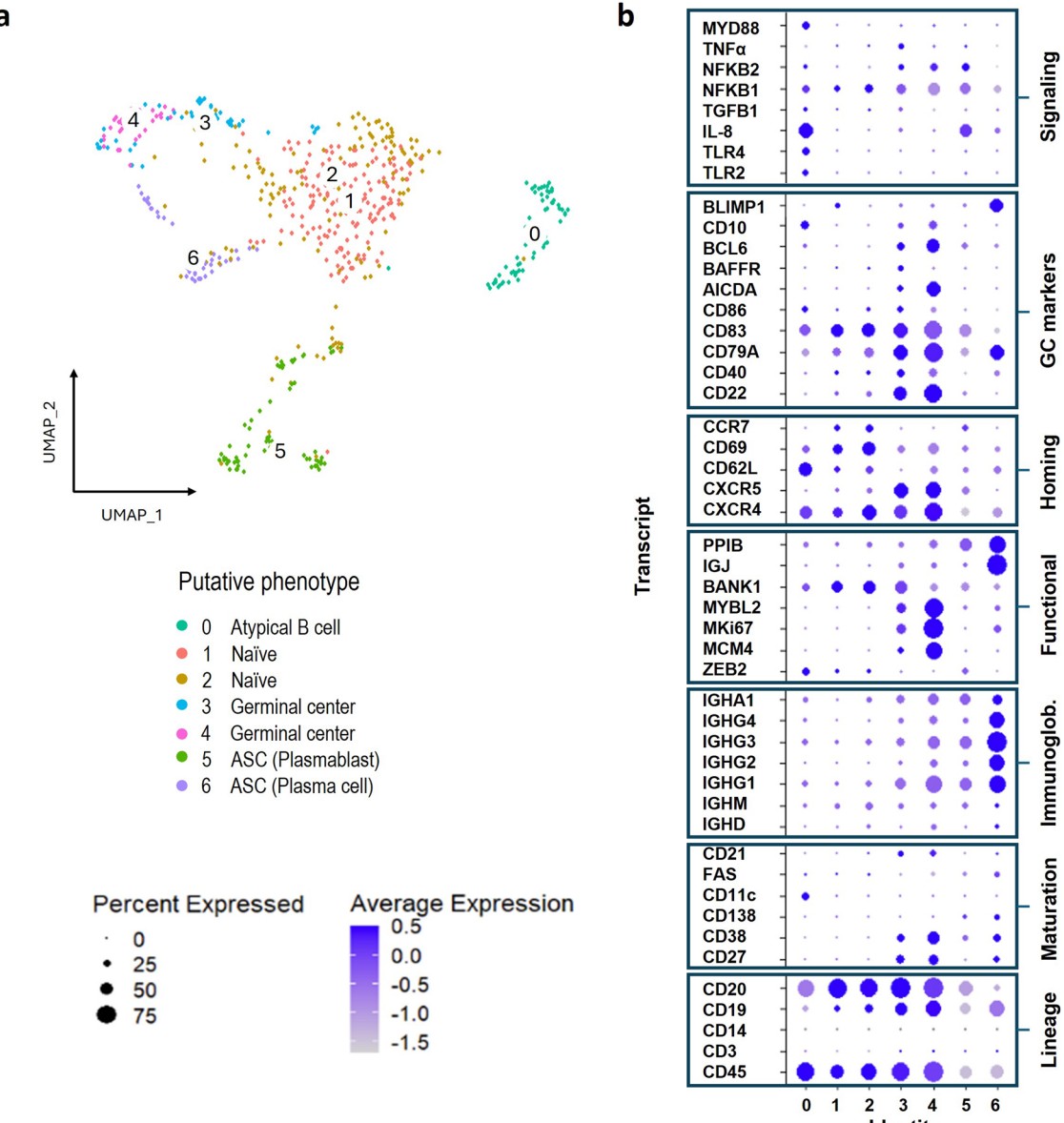

**Fig. 4 | Differential gene expression identifies subclusters amongst B cells isolated from human lung tissue from patients with TB.** scRNA sequencing was used to explore the differentially expressed genes in B cells isolated from human lung tissue from patients with TB (*n* = 9). **a** B cells were clustered based on their differential gene expression and visualised by UMAP with a total of seven putative clusters identified (0 to 6). **b** These B cell clusters were interogated for their expression of specific B cell-associated genes to delineate their likely functions.

involved in the production of IgM and IgA antibodies[39–42] and protein secretion, respectively[43]. Consistent with flow cytometry data, lung B cells express the gene encoding CD69, especially subsets 1 and 2, which also express CCR7 and CXCR4. Subsets 3 and 4 lack CCR7 but co-express CXCR4 and CXCR5. All three chemokine receptors are involved in trafficking within the germinal centre (GC)[7,39,44], and may thus represent B cell populations associated with GrALT. In line with this, populations 3 and 4 express Bcl6, a key regulator of the GC response[45–47] and AICDA, which is expressed during B cell receptor (BCR) affinity maturation[48], as well as the GC marker CD10[35]. Population 3 and 4 also express the highest levels of CD21 and CD22, both essential to B cell GC survival[36,49] and CD79a, a marker of mature B cells[9]. The immunoglobulin and CD79a transcript abundance generally increased from populations 0 to 6, and since both are associated with the B cell surface receptor, this also suggests the local B cell population is maturing[9,50]. Several genes associated with genome replication (MCM4)[45], proliferation (MKi67)[42,45] and cell cycle progression (MYBL2)[46] are also highly expressed. Population 0, on the other hand, is distinguished by CD11c expression, which

is a marker of atypical B cells[51–54], and ZEB2, recently identified as a key transcription factor promoting the development of this cell type[55]. This population also expresses genes encoding TLR2, TLR4 and MyD88, a common adaptor molecule for all TLRs except TLR3[56]. Interestingly, BANK-1 was recently shown to signal together with MyD88 and TLRs to co-ordinate innate immune signalling B cells, including the production of IL-8, and population 0 expresses both BANK-1 and high levels of the IL-8 gene[57]. Thus, scRNAseq data supports the presence of GC-like B cell populations, ASC-like and atypical/innate-like B cells in lung tissue from patients with TB. The next aim was to identify similarities between these B cell populations putatively identified by sequencing, with those identified by flow cytometry.

**Transcriptional heterogeneity of B cells translates to several B cell phenotypes identified by flow cytometry, highlighting compartmental heterogeneity**

We obtained 13 additional lung samples and analysed lung homogenate and matched blood samples by flow cytometry using three separate antibody

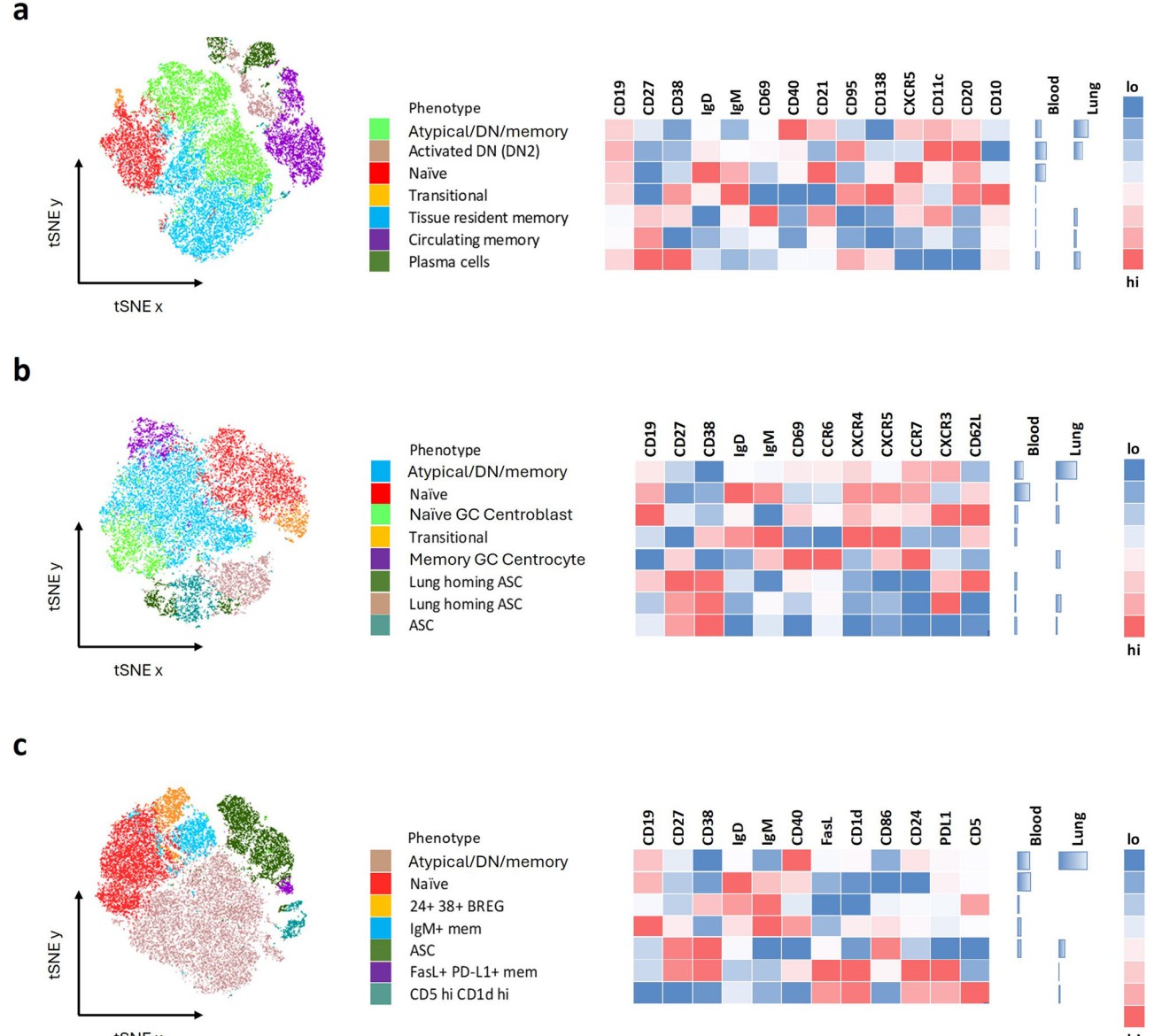

**Fig. 5 | tSNE plot comparison of B cell phenotypes further refines blood and lung tissue associated compartments.** In order to study the complexity of the B cell phenotypes associated with blood and lung tissue compartments of patients with TB, three separate flow cytometry phenotyping panels were used. **a** Assesed the level of maturation of the B cells, **b** the expression of different homing markers and **c** the relative expression of regulatory markers. The different B cell populations were clustered spatially using tSNE, with a single patient presented here, followed by a key describing the major phenotypes each of which is assigned a different colour. The relative expression of the specific markers associated with each phenotype were plotted alongside in a heatmap. Since the blood and lung samples were concatenated into a single file, the relative frequencies of cells in the blood or lung tissue compartment could be compared as shown alongside the heatmap.

panels, each elaborating on a core set of shared markers included in Figs. 1 and 3: CD19, CD27, CD38, IgD and IgM (Fig. 5). We initially analysed these data using an unbiased clustering algorithm to identify unique B cell populations based on the markers in each flow cytometry panel (Fig. 5a–c). Consistent with the initial flow cytometry profiling (Fig. 3), the main phenotypic differences were based on B cell maturation, with the blood containing more transitional and naive B cells, compared to increased memory and ASC frequencies in the lung. A single patient sample was used to generate a representative tSNE plot (Fig. 5), to gain an overview of the number of unique B cell phenotypes associated with the lung and blood compartments. We then interrogated any phenotypes of interest in a total of 13 patients with TB to assess compartmental differences (Fig. 6, Supplementary Fig. 3).

Common to all three panels was a prominent B cell phenotype that was CD27[mid/lo], CD38-, IgD- and IgM- (Fig. 5a green, b blue, c tan) designated as an atypical/double negative (DN) memory population. The double negative phenotype is defined as being negative for CD27 and IgD[51]. This was expanded in the lung compared to matched blood. As shown in flow panel 1 (Fig. 5a), two such populations were apparent, both of which express the key atypical B cell marker CD11c, consistent with the scRNAseq data (Fig. 4). The second population resembles an activated double negative (DN2) phenotype, as it expresses the highest levels of CD11c, and is CD20[hi] and CXCR5-[51–54]. This activated DN2 phenotype was enriched in the lung when assessed across the 13 patients (Fig. 6a, Supplementary Fig. 3a). Several studies have observed an enrichment of DN2 B cells in the blood under inflammatory conditions[51–54] and further enrichment in the lung of patients with TB supports their potential importance at the site of infection. In addition, in the second panel, the putative atypical B cell population (Fig. 5b blue) expresses high levels of CXCR3, which would facilitate homing to inflamed tissue sites via CXCL10[27,39,58].

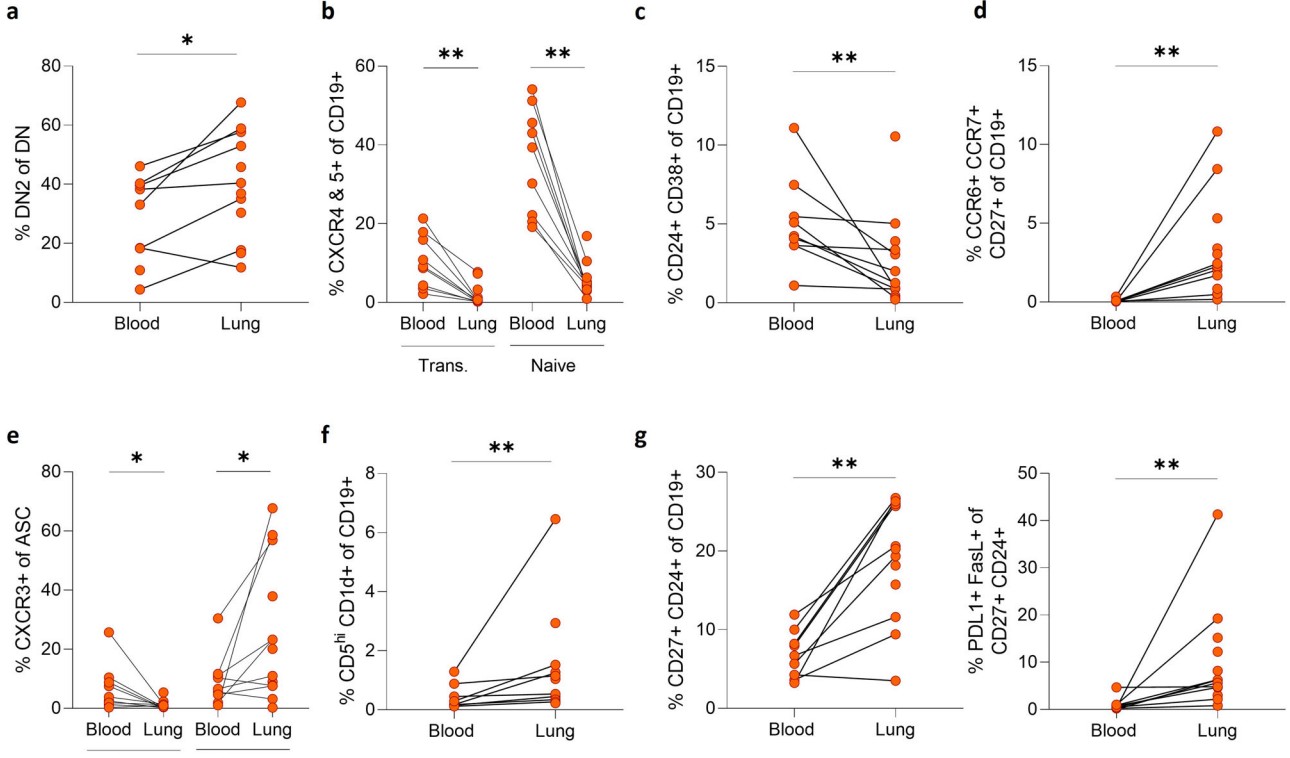

**Fig. 6 | tSNE-derived unique B cell phenotypes associated with the blood or lung compartment of patients with TB. a** Double negative (DN) B cells (CD27⁻IgD⁻) were assessed and the activated (DN2) phenotype (CD21⁻CD11c⁺) was enriched in the lungs of patients with TB. **b** GC homing (CXCR4⁺CXCR5⁺) transitional and naive B cells were enriched in the blood compartment. **c** The blood compartment was enriched for a tranistional regulatory (CD24⁺CD38⁺) population. **d** The lung compartment contained a unique activated B cell memory (CD27⁺CXCR5⁺CD62L⁻CD69⁺CCR6⁺CCR7⁺) population. **e** Lung-derived ASC populations expressed a CXCR3⁺CD62L⁻ phenotype relative to the blood-derived counterparts. Other B cell phenotypes associated with regulatory functions were enriched in the lung, including **f** a CD5⁺ phenotype (CD5hiCD40⁻CD1d⁺) and **g** a CD27⁺ regulatory population (CD27⁺CD24⁺FasL⁺PDL1⁺). A set of 13 samples from patients with TB were used in these analyses. Statistical analyses were performed using the Wilcoxon test for matched/paired samples. *P* values are denoted by * ≤0.05 and ** <0.01.

Naive (Fig. 5, red) and transitional (Fig. 5, orange) B cell phenotypes were dominant in the blood, as seen in the initial flow cytometry phenotyping data (Fig. 3), with the transitional phenotype expressing CD10 and CD138 as expected (Fig. 5a)[37]. Both naive and transitional phenotypes displayed high expression of CXCR5 associated with homing to germinal centres (Figs. 5b and 6b, Supplementary Fig. 3b)[7,39,44], and both phenotypes were enriched in circulation (Fig. 6B). The naive GC centroblast-like phenotype (Fig. 5b, lime green) expressed homing markers including CD62L, CXCR4, CXCR5 and CCR7, consistent with population 1 and 2 identified by scRNAseq (Fig. 4). A transitional B cell phenotype, CD24⁺ CD38⁺, which is associated with regulatory functions[42,59], was enriched in the blood compared to lung (Figs. 5c and 6c, Supplementary Fig. 3c). In contrast, the purple population in Fig. 5b, which expressed a phenotype resembling the putative GC populations 3 and 4 from the scRNAseq data (Fig. 4), being CD27⁺CXCR5⁺CD62L⁻CD69⁺CCR7⁺ and CCR6⁺, was significantly enriched in the lung (Fig. 6d, Supplementary Fig. 3d).

The CD27⁺CD38hi ASC-like populations observed by scRNAseq (Fig. 4) were also apparent by flow cytometry in the lung, including CD138⁺ plasma cells (Fig. 5a). ASCs could be separated based on expression of CXCR3 and CD62L (Fig. 5b). CXCR3 facilitates homing to sites of inflammation, and CD62L (L-selectin) is essential for migration of lymphocytes into tissue and is lost as B cells mature into plasmablasts and finally plasma cells[60]. CXCR3⁺CD62L⁺ ASC were significantly enriched in the blood, whereas CXCR3⁺CD62L⁻ ASC were enriched in the lung (Figs. 5b and 6e and Supplementary Fig. 3e), consistent with antibody-secreting populations (5 and 6) observed by sequencing (Fig. 4). In addition, two phenotypes associated with regulatory functions were significantly associated with the lung environment, one expressing a CD5⁺CD1d⁺ phenotype

(Figs. 5c and 6f and Supplementary Fig. 3f)[26,61] and the other a memory CD27⁺CD24⁺ phenotype[59] (Fig. 6g and Supplementary Fig. 3g). The second putative regulatory population also expressed PDL-1, the receptor for PD-1, important for regulating the T cell response in TB[3,62,63]. The importance of antigen-specific memory B cells and the PDL-1/PD-1 axis for localising T_FH cells within GrALT and mediating TB control in both mice and macaques was recently demonstrated[3]. Together these flow cytometry data confirm the presence of diverse B cell phenotypes in lung tissue from patients with TB, including an expanded CD11c expressing atypical B cell population, subsets potentially associated with GC activity, antibody production, and regulation.

## B cells impact *Mtb* growth in a 3D granuloma biomimetic model

To investigate the potential effect of B cells on *Mtb* growth, we turned to a 3D biomimetic culture model that resembles the granuloma microenvironment[64,65]. The model uses peripheral blood mononuclear cells (PBMC) infected with bioluminescent *Mtb*. These *Mtb*-infected PBMC are then encapsulated in collagen/alginate microspheres that contain an extracellular matrix scaffold. Each microsphere therefore represents a 3D microenvironment that mimics an individual granuloma. Without B cell depletion, the overall *Mtb* growth rate was significantly greater in undepleted PBMC from patients with TB compared to control samples, consistent with immune dysregulation during TB disease (Fig. 7a). CD19⁺ B cells were depleted from PBMC using positive magnetic selection and *Mtb* growth kinetics measured compared to non-depleted microspheres (Fig. 7b). Using PBMC from four healthy donors, with each condition performed in triplicate, demonstrated that B cell depletion caused a consistent and significant increase in *Mtb* growth (Fig. 7c, d). Interestingly

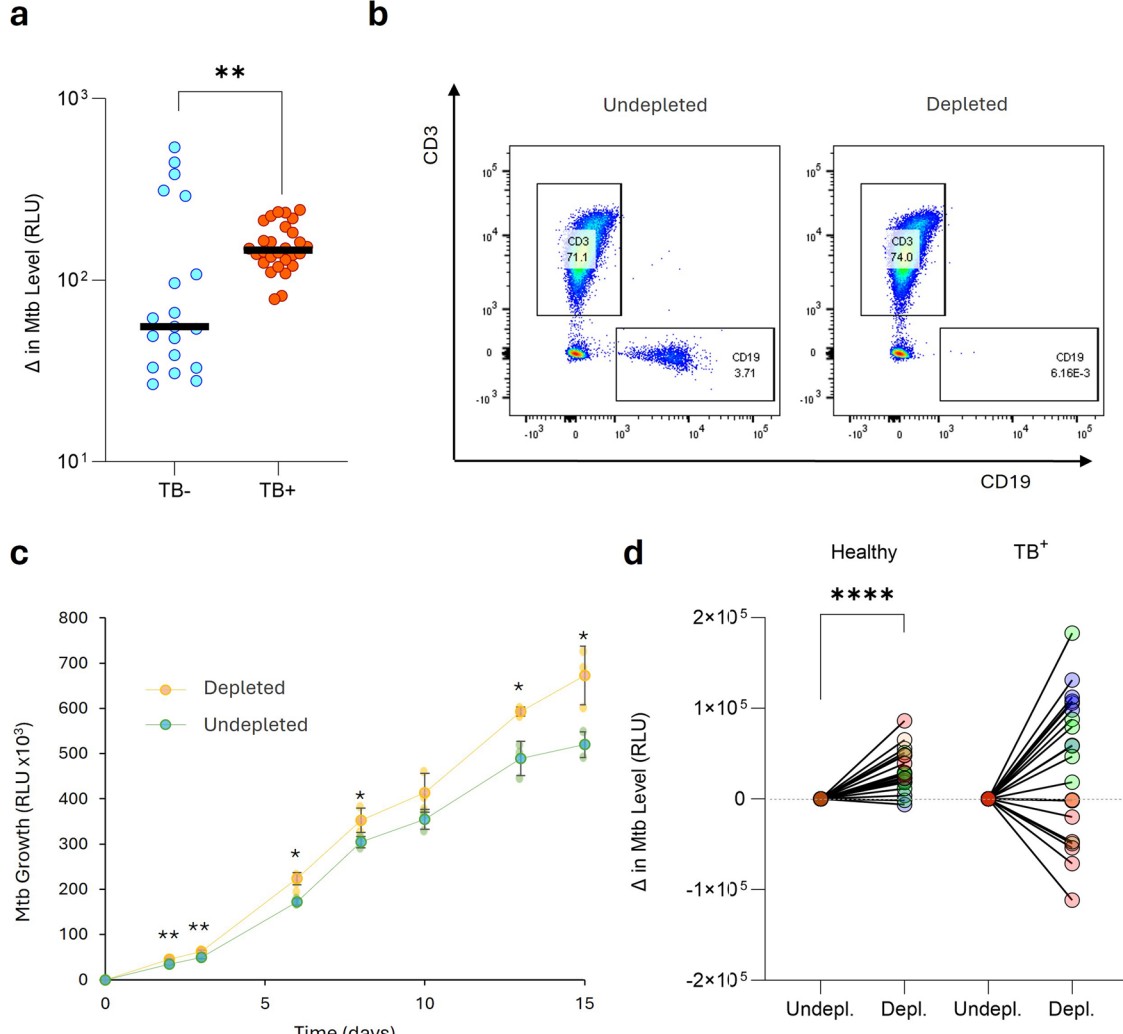

**Fig. 7 | B cells contribute to host control of *Mtb* in healthy donors in a 3D biomimetic model, but in patients with TB the effect is highly variable.**
**a** Comparatively, *Mtb* grew more rapidly in PBMC derived from patients with TB. B cells were depleted from PBMCs, of which four donors were healthy and four had active TB, and compared to undepleted PBMCs. **b** Representative FACS plots of the depletion process. **c** B cell depletion leads to significantly higher *Mtb* growth in a

healthy donor PBMC sample, representative of four healthy donors. **d** *Mtb* outgrowth data at day 7 from a total of four PBMC donors from each group are summarized, demonstrating the highly variable effect of B cell depletion in patients with TB. Data were normalized to the undepleted control group and compared by Wilcoxon matched-pairs signed rank test. Standard deviation is presented in (**c**). *P* values are denoted by * ≤0.05; ** <0.01 and **** <0.0001.

though, the effect of depletion on PBMC from patients with active TB prior to the initiation of drug therapy was highly variable, with *Mtb* growth increasing in two of the donors and decreasing in the remaining two (Fig. 7d). Together, these data demonstrate that B cells can regulate *Mtb* growth, but this effect is variable in active TB suggesting their function can be altered by ongoing disease.

**Lung tissue derived antibodies show *Mtb* specificity and enhance bacterial phagocytosis**
Having observed an expansion of ASC subsets in lung tissue from patients with TB, we compared the relative frequency of ASCs of TB and non-TB patients and found an enrichment in lung tissue in both groups (Fig. 8a). The ASCs expressed the tissue residence marker CD69 and the plasma cell marker CD138[27,37]. These compartmental differences prompted us to investigate the antibody profiles in lungs from patients with TB. Therefore, we isolated antibodies from lung tissue homogenates using a thiophilic gradient (Fig. 8b)[66] and confirmed the presence of IgM, IgD, IgG and IgA (Fig. 8c; IgE was not detected). The purified antibodies were reactive to whole *Mtb* lysate by ELISA, and significantly enriched in the patients with TB (Fig. 8d). Although individual *Mtb*-reactive antibody classes showed

similar upregulation, only IgM was significantly upregulated in the patients with TB (Fig. 8e). Interestingly in this highly TB endemic population, there was no difference in the reactivity of plasma-derived antibodies from the same TB and non-TB diseased participants (Supplementary Fig. 4). These data indicate the enrichment of TB-specific antibodies in TB diseased human lung tissue, supporting a role at the site of disease.

Finally, to confirm the specificity of these antibodies and to test their functionality, purified antibodies were studied using the 3D biomimetic *Mtb* culture model[65]. Antibodies purified from lung homogenate of patients with TB enhanced the phagocytosis of *Mtb*, shown by increased uptake of luminescent *Mtb* compared to isotype controls (Fig. 8f). Although the flow cytometry does not determine if the bacteria are phagocytosed or cell-associated, the cells are extensively washed post-infection, and we have previously shown that *Mtb* growth in this system is predominantly intracellular[64]. Subsequent growth of *Mtb* over the next two weeks was markedly greater when *Mtb* had been treated with these antibodies (Fig. 8g). However, when the results were normalized to the initial infectious load, no significant change in the growth rate was observed, suggesting the antibodies do not modulate subsequent proliferation, at least within this model system (Fig. 8h). Thus, these data confirm the presence of *Mtb* specific

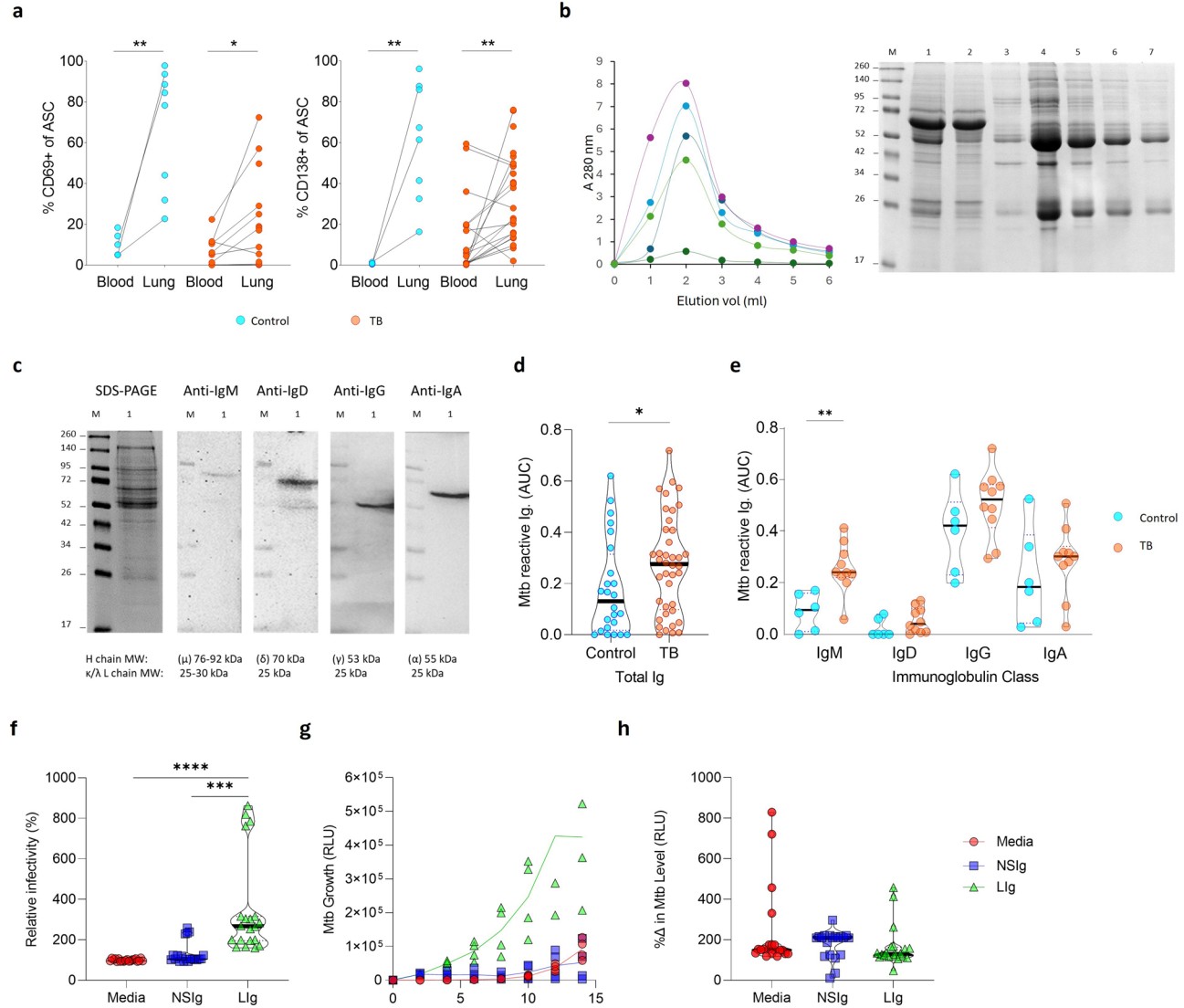

**Fig. 8 | Mature CD138⁺ plasma cells are enriched in the lung tissue, resulting in an upregulated *Mtb* specific antibody response that enhances *Mtb* phagocytosis.** **a** The tissue residence marker CD69 was expressed on antibody-secreting cells (ASC), with the majority displaying a mature plasma cell CD138⁺ phenotype in both cancer controls ($n = 8$) and patients with TB ($n = 13$ to 25). **b** Antibodies were isolated from homogenised lung tissue supernatants using a thiophilic resin, with example elution profiles from five lungs measured at 280 nm, and then examined on a reducing 12.5% SDS-PAGE gel. The molecular weight marker (lane M), was followed by the initial lung homogenate supernatant sample (lane 1), the unbound sample (lane 2) followed by the eluted fractions 1–5 from the elution profile alongside (lanes 3–7). **c** Peak eluents (lane 1) were tested for the presence of IgM, IgD, IgG and IgA using class-specific secondary antibodies by western blot. **d** *Mtb*-specific antibody binding was investigated in lung samples from cancer controls ($n = 6$) and patients with TB ($n = 10$), detecting both total reactivity and **e** class-specific responses in a direct ELISA-based approach, expressed as area under the curve (AUC). An equal concentration range (100 to 0.8 µg/ml) of isolated antibody

was used to standardize the assay and allow for comparison between patients. Antibody frequencies were compared by Mann–Whitney test. **f** *Mtb* phagocytosis by PBMCs was compared between untreated *Mtb* versus *Mtb* treated with non-specific immunoglobulin (NSIg) or lung immunoglobulin (LIg) from patients with TB. The lung-derived immunoglobulin increased phagocytosis, while non-specific immunoglobulin had no effect. Data are presented as summary data from five immunoglobulin samples from patients with TB, tested with three different PBMC donors, with each experiment done in quadruplicate and infectivity was measured by luminescence. Data were normalized to the media control and the donor matched mean differences were compared by one-way ANOVA. **g** Representative *Mtb* growth using immunoglobulin from patients with TB comparing with untreated and non-specific immunoglobulin in the 3D biomimetic model of TB. **h** *Mtb* luminescence at day 8 normalized to day 0 initial infectious load shows no overall difference in growth rate over time. *P* values are denoted by * ≤0.05; ** <0.01; *** <0.001 and **** <0.0001.

antibodies in TB affected lung tissue, which enhance the phagocytosis of mycobacteria.

## Discussion

Diverse lines of evidence, from human studies and animal models, demonstrate that B cells are involved in the host response to *Mtb*, but B cells represent a multi-faceted immune subset, and their precise role has remained elusive. The frequency of B cells in circulation is reduced during

active TB[11,13–17] and infection is known to induce B cell containing tertiary lymphoid follicles in the lung referred to as granuloma-associated lymphoid tissue (GrALT)[3,8,11,12]. Using matched blood and lung samples, we show that these phenomena overlap, with decreased B cells in the blood reflecting an increase in the lungs of patients with TB. Lung B cell frequency was extremely high in some participants, and, by histology, they were found to be associated with the granuloma in similar quantities to T cells. We next performed a detailed characterization of B cell phenotypes in human TB

lung tissue using complementary scRNAseq and flow cytometry approaches and interrogate the potential functional impact in a cellular model. For technical reasons, it was not possible to perform scRNAseq analysis and FACS analysis on the same samples, however, the fact that we observe similar changes at both the transcriptional and protein level supports the phenotypic changes observed by each approach, despite analysis being in different samples.

A central focus of our study was to compare B cells in the lung compartment with those in circulation, which have been the focus of most studies in humans. In contrast to a predominantly naive and transitional phenotype in the blood, lung B cells were mostly of a memory and ASC phenotype and expressed the tissue residence marker CD69, consistent with resident memory cells[67]. There is limited data regarding human lung derived B cell phenotypes, but our data are supported by Barker et al. and Tan et al., who also observed CD69+ expression[68,69]. Mechanistically, from mouse models, the recruitment of B cells to the lung requires local antigen encounter[27,69,70]. The initial seeding of memory B cells in the lung is derived from the mediastinal lymph node[69], with lung resident cells expressing CXCR3, CCR6 and CD69[27,69]. These cells patrol the lung and upon local antigen encounter either form part of induced bronchus-associated lymphoid tissue (iBALT), or GrALT in the case of TB, or differentiate into ASC[27,70]. We observed memory B cells and ASCs expressing all three markers, supporting their tissue residence. Interestingly, Tan et al. detected a resting memory B cell phenotype and little expression of GC-associated markers Bcl6, Ki67 and AICDA in healthy lungs[69]. In contrast, we observed two memory B cell populations expressing these transcripts in our scRNAseq data, in addition to a memory B cell phenotype expressing both CCR6 and CCR7 homing markers by flow cytometry. We also detected both transcripts and surface expression of the plasma cell marker CD138. Together, our findings reflect the diseased state of these lungs from patients with TB, suggestive of local antigen encounter by B cells in the lung and their involvement in GrALT, with differentiation into ASCs and hence a tissue-resident B cell response.

Several of our findings point toward the possible function or interaction of B cells with T cells within the GrALT. CCR7 facilitates migration of B cells toward the T cell zone within GC, allowing for B cell help from T$_F$H cells[44,71] and a key interaction with T$_F$H cells includes the CD40-CD40L interaction[72], all of which were expressed by lung B cells. Linge et al. demonstrate that B cells are an important source of IL6 and that the T$_H$1 and T$_H$17 response[23] is affected when IL6 is knocked out of B cells[21]. Regarding putative regulatory B cells observed in the lung, recent evidence suggests that antigen experienced memory B cells expressing PDL-1 are important in recruiting T$_F$H cells into GC[3]. Moreover, Zhang et al. show that CD5$^{hi}$CD1d+ B cells can inhibit IL-17 production by T cells[26]. These cells were present in lung tissue from patients with TB and have previously been observed in circulation of patients with ATBI and expanded in those with cavitation[25,32]. The local inflamed milieu of the lung is also reflected by the presence of atypical B cells, which are associated with an inflamed state[51,53]. The atypical cells expressed TLR2 and 4, of which the former could detect bacterial lipoprotein such as ManLAM for example, resulting in MyD88 signalling and the IL8 response, which could result in the local recruitment of immune cells including T cells[73,74]. Importantly, CD11c B cells have been associated with many inflammatory conditions in humans including granulomatous lung diseases such as sarcoidosis[75]. In that study, these were termed age-associated B cells (ABCs), which share the same markers and probably represent the same or very similar B cell populations[76]. In addition, atypical or ABC are associated with many autoimmune conditions including Lupus and multiple sclerosis, and TB shares clinical similarities with such autoimmune conditions[77,78]. It is possible, therefore, that the atypical B cells enriched in lungs from patients with TB play a role in the immunopathogenesis of TB, through mechanisms including antibody and cytokine production and cross talk with T cells. Overall, the heterogeneity of B cells observed in lung tissue from patients with TB is consistent with multiple putative roles in TB protection and

pathogenesis. We do concede that these data are only descriptive of the array of B cells found in the human lung with TB, and specific functional roles remain to be mechanistically proven.

To investigate the effect of B cells on *Mtb* in the lung, we used a granuloma biomimetic model and found that depletion of B cells increases *Mtb* growth. This is consistent with mouse[6,7] and NHP[10] studies showing an increased pulmonary bacterial burden in the absence of B cells. In addition, Joosten et al. observed a negative correlation between B cell frequency and bacterial growth in humans, using a whole blood mycobacterial growth inhibition assay[79]. Consequently, all these studies support a protective role for B cells, but notably in our model this control is inconsistent when PBMCs from patients with active TB were used. We believe it is likely to reflect the fact that the patients with TB were, by chance, at very different stages of disease severity. For example, in more severe TB disease we expect higher frequencies of pro-inflammatory atypical B cells, which may contribute to *Mtb* growth. The depletion of B cells in these cases may reduce *Mtb* growth, both in the 3D model and by analogy, in the lung. This is supported by the fact that *Mtb* growth was higher in PBMCs from patients with TB vs controls before B cell depletion (Fig. 7a). However, additional experiments are required to explore this hypothesis further. This suggests that B cells can contribute to control of *Mtb* growth, but also that the B cell compartment is disrupted during active TB. In a longitudinal study, Moreira-Teixeira et al. observed a decreased B cell signature with increased disease progression in the blood of patients with LTBI, LTBI-progressors and patients with ATBI, consistent with our observation[16]. Similarly, several studies showed that the proportions of B cell populations and transcriptional signatures are affected by TB[11,15,16,80].

Finally, we analysed the antibodies present in TB-affected lung tissue and found significantly increased levels of those specific for *Mtb*. This was observed for all antibody classes, but only reached significance for IgM. Interestingly, IgM is known to be a potent activator of the complement system, which is a strong biomarker of TB disease[81]. In addition, IgM memory B cells are associated with induced lymphoid tissue in mucosal barriers[82], suggesting a potential link with the GrALT observed in TB affected lungs. IgM memory B cells are emerging as important effectors of both adaptive and innate-like immune responses in mucosal barriers[82]. Antibodies from the lungs of infected patients enhanced uptake of *Mtb*, confirming both specificity and consistent with opsonisation increasing phagocytosis of mycobacteria[83]. However, we did not detect an effect on subsequent *Mtb* growth in our granuloma biomimetic model. A strong IgM response was associated with protection against TB disease in NHPs vaccinated with BCG via an intravenous route[84]. In addition, studies in humans have shown enhanced phagocytosis, phagolysosome fusion and macrophage killing using plasma antibodies isolated from TB resistors compared to those with active TB disease[29,85]. As lung antibodies used in this study were isolated from subjects with advanced TB lung disease, it is therefore perhaps expected that they failed to reduce *Mtb* growth despite enhanced phagocytosis. However, the fact that TB-specific antibodies are present at the site of disease in humans supports a role in the immune response to *Mtb* infection in humans.

B cell phenotype and function differs significantly between the lung and circulation, and B cells can regulate the host-pathogen interaction in TB. B cells isolated from human lung tissue displayed unique phenotypic subsets including populations consistent with tissue-resident memory, GC activity, atypical and regulatory B cells, suggesting potentially diverse roles in the immune response to *Mtb*. In addition, *Mtb*-specific antibodies are enriched in lung tissue from patients with TB and augment phagocytosis of *Mtb*. These data show B cells are likely to contribute to the host response to TB in humans at multiple points and highlight that considering tissue-specific phenotypes is likely to be essential to fully understand their overall impact. Future studies will aim to assess the TB specificity of the B cells in the GrALT and focus on spatial transcriptomic approaches to investigate the nature of the B cell follicles within the lung and if they form part of protection or pathology.

## Table 1 | Descriptive table of the study cohort (*n* = 76)

|        | Gender   | + Lymph node | HIV +ve   | TB        | Smokers  | Age[a]     |
|--------|----------|--------------|-----------|-----------|----------|------------|
| Male   | 47 (62%) | 7            | 20 (43%)  | 34 (72%)  | 21 (45%) | 42 (35-48) |
| Female | 29 (38%) | 10           | 19 (66%)  | 24 (83%)  | 1 (3%)   | 39 (34,47) |
| **Total** | **76** | **17**      | **39 (51%)** | **58 (76%)** | **22 (29%)** |    |

The total sample numbers and percentages in each column are highlighted in bold.
[a]Median and IQR are indicated.

## Methods

### Cohort description
This study made use of a cohort of patients that underwent thoracotomy or lung resection surgeries. This included patients with a history of TB and non-TB-associated pathologies, with or without HIV. Written informed consent was obtained from all enrolled participants and surgeries were performed at either the King Dini-Zulu Hospital Complex or Inkosi Albert Luthuli Hospital in Durban, KwaZulu-Natal, South Africa. Patients were assessed for the extent of lung disease (cavitation and/or bronchiectasis) via HRCT. Candidates for surgery had to pass a fitness test as determined by Karnofsky score, 6-min walk test, spirometry, and arterial blood gas. Patients with massive hemoptysis were further assessed for their general condition, effort tolerance prior to hemoptysis, arterial blood gas measurement, serum albumin level and HRCT imaging. Tissues were collected following surgery to remove the irreversibly damaged lobes or lungs. Resected lung tissue was dissected into smaller tissue samples on site and transferred to RPMI 1640 media containing 10% FBS and kept on ice. The sections selected for analysis represented visibly low-, mid- and severely diseased areas of the lung, which were combined and processed[86]. Therefore, the isolated cells were representative of the entire lung and were not necessarily only of granulomatous origin. Samples were received for processing within 1 h of dissection and processed immediately in a BSL3 facility.

The cohort consisted largely of a Black African population 82% (90% of males and 74% of females). A total of 76 lung samples were collected, including 17 lung-draining lymph nodes and all samples with matched blood (Table 1). About half of the cohort was HIV positive, with positivity skewing slightly toward females. Most resections were TB-related (76%), with a median age of 41 years, and a relatively similar age distribution amongst males and females. Close to two-thirds of cases were males, and of these 45% were active/previous smokers. Lung cancer resections were dissected, and the healthy tissue margins were retained as the TB-negative control lung samples[86].

### Histology
A section of lung was cut and transferred to 10% buffered formalin to fix. The sample was then processed in a vacuum filtration processor using a xylene-free method and isopropanol as the main substitute fixative. The tissues were embedded in paraffin wax and then cut into 4 μm sections, baked at 60 °C for 15 min, dewaxed using two xylene changes and rehydrated with descending grades of alcohol to water. These were then Hematoxylin and Eosin (H&E) stained using standard procedures, then dehydrated again in ascending grades of alcohol, cleared in xylene and mounted with a distyrene, plasticizer and xylene solution (DPX). The H&E slides were used to assess specimen quality before proceeding to immunohistochemistry.

Tissue sections (4 μm) of specimens of good quality were mounted on charged slides, and heated at 56 °C for 15 min. Sections were dewaxed in xylene, rinsed in 100% ethanol and one change of SVR (95%), washed under running water for 2 min followed by antigen retrieval via Heat Induced Epitope Retrieval (HIER) in Tris-HCl (pH 6.0) for 30 min. Slides were cooled for 15 min and rinsed under running water for 2 min. Endogenous peroxidase activity was blocked using 3% $H_2O_2$ for 10 min at room temperature (RT), followed by a wash in PBST and blocking with protein block (Novolink) for 5 min at RT. Slides were incubated with primary antibodies for CD20 (M0755-CD20cy-L26, DAKO), CD68 (ab192847, Abcam), CD45

(M0701-2B11 + PD7/26, DAKO), CD3 (ab16669, Abcam) and CD21 (M0784-1F8, Dako), followed by washing and incubation with the polymer (Novolink) for 30 min at RT. Finally, the slides were washed and stained with diaminobenzidine (DAB) for 5 min, washed and counterstained with hematoxylin for 2 min, washed and blued in 3% ammoniated water for 30 s, washed, dehydrated, and mounted in DPX. Slides were imaged using a Hamamatsu NDP slide scanner (Hamamatsu NanoZoomer RS2, Model C10730-12) and its viewing software (NDP.View2). The red, green, and blue colour balance was kept at 100% whereas gamma correction was maintained between 0.7 and 2. Brightness (60–110%) and contrast (100–180%) settings varied slightly between slides depending on staining quality. Resolution was 230 nm/pixel yielding file sizes of 2–4.4 GB. Contrast, brightness, and intensity of exported images (jpg format) were minimally adjusted using CorelDraw 2020. DAB-stained cells were quantified using QuPath (https://qupath.github.io/) with the operator blinded to the nature of the sample.

### Peripheral blood mononuclear cell (PBMC) isolation
Blood samples were collected in EDTA tubes and processed within 4 h of collection. Samples were centrifuged $930 \times g$ for 5 min and 1 ml plasma fractions were aliquoted and stored at −80 °C. The remaining red blood cell pellet was made up to three times its pellet volume with PBS warmed to room temperature. PBMCs were isolated using a Ficoll-Paque (Amersham Biosciences, Little Chalfont, UK) density gradient sedimentation method as per manufacturer's instructions. Isolated PBMCs were cryopreserved in 10% DMSO in heat inactivated foetal bovine serum (FBS, ThermoFischer Scientific) and stored in liquid nitrogen until use.

### Lung mononuclear cell (LMC) isolation
Tissue specimens were mechanically dissociated using scissors followed by GentleMACs (Miltenyi Biotec) homogenization for 15 s in RPMI 1640, 10% FBS, 40 μg collagenase D (Roche) and 40 U/ml DNASe I (SIGMA-Aldrich). Samples were then incubated for 30 min at 37 °C, followed by another 75 s homogenization cycle and passed through 70 μm cell strainer (Corning). Following a 5 min $930 \times g$ centrifugation (Beckman Coulter, Allegra X-12R), the sample pellet was suspended in 5 ml and passed through a 40 μm cell strainer and centrifuged as before. Finally, the pellet was treated with 5 ml red blood cell lysis solution (QIAGEN) for 5 min, made up to 30 ml with PBS and centrifuged again. Cells were counted and split into equal numbers (1–5 million) to stain for flow cytometry. Lung draining lymph nodes were processed similarly without the need for GentleMACs homogenization or collagenase D and DNAse I treatment.

### Flow cytometry
Isolated LMCs were processed for flow cytometry on the same day to avoid reduced viability following cryopreservation and thawing. To facilitate running matched blood samples on the same day as stained LMCs, isolated PBMCs were thawed on the day the lung samples were processed. Briefly, cryopreserved PBMCs were thawed, washed, and rested in RPMI 1640 containing 10% FBS for 1 h in a 37 °C, 5% $CO_2$ incubator prior to staining. Between 1–5 million PBMCs or LMCs were stained with the respective B cell surface marker panels (Supplementary Tables 1 to 3) for 20 min at RT in the dark. Samples were washed twice with PBS and suspended in 250 μl 2% PFA-PBS and kept at 4 °C in the dark. Samples were acquired on a BD FACS Aria Fusion III and data analysed using FlowJo version 9.9.6 (Tree Star).

### Lung homogenate and plasma antibody isolation
All homogenate supernatants throughout the LMC isolation protocol were reserved for isolation of patient lung-derived immunoglobulins. The homogenate supernatants were centrifuged at 10,000 RCF for 10 min to pellet any remaining cellular debris. These were then filtered through a 0.22 μm syringe filter (Millipore, Merck). This clarified supernatant was then incubated with a Thiophilic resin (Pierce, ThermoFischer) for isolation of immunoglobulins[66]. For plasma antibodies, 2 ml plasma was loaded onto a thiophilic resin. Each sample type had a designated thiophilic column to avoid cross contamination of purified immunoglobulin pools. Both sample

types were incubated with the resin for 1 h at room temperature, after which the bound immunoglobulins were eluted as per the manufacturer's instructions. The resulting eluted sample absorbance was measured at 280 nm and fractions of interest were run on SDS-PAGE and western blot to assess sample purity, after which they were pooled by sample type. Sterile glycerol was added to 50% (v/v) and the pooled samples were stored at −20 °C.

## SDS-PAGE and western blot of isolated immunoglobulins

Isolated immunoglobulin fractions were prepared for analysis on 12.5% reducing SDS-PAGE gels[87]. Samples were prepared in an equal volume of reducing SDS-PAGE sample buffer containing 10% (v/v) β-mercaptoethanol and boiled for 5 min before loading on duplicate SDS-PAGE gels. Once completed, one gel was stained with Coomassie brilliant blue R250 (SIGMA) as a reference gel, while the second was transferred (Trans-Blot Turbo, Bio-Rad) to nitrocellulose membrane for western blotting. PBS with 0.1% Tween-20 (PBST) was used throughout for western blotting. Membranes were blocked with 5% low-fat milk powder in PBST for 1 h, washed twice with PBST and incubated with immunoglobulin class-specific primary antibodies (all at 1:5000 dilution) for 2 h: Goat anti-human IgD-HRPO (cat. no 2030-05, Southern Biotech), Goat anti-human IgA-HRPO (cat. no 2050-05, Southern Biotech), Donkey anti-human IgG-HRPO (cat. no 709-036-073, Jackson ImmunoResearch), Donkey anti-human IgM-HRPO (cat. no 709-036-098, Jackson ImmunoResearch). Blots were washed again with PBST followed by a final wash and detection with enhanced chemiluminescence substrate (Pierce, ThermoFischer) using a ChemiDoc™ imager (Bio-Rad).

## ELISA to test *Mtb* specificity of isolated immunoglobulins

*Mycobacterium tuberculosis* (*Mtb*) H37Rv cultures were used to prepare a lysate for ELISAs. The *Mtb* lysate protein concentration was determined by Pierce 660 nm protein assay (Pierce, ThermoFischer) and used to coat ELISA plates at 10 μg/ml of lysate in PBS, 100 μl/well overnight at 4 °C. Plates were washed three times between each incubation step, using 200 μl/well PBS with 0.1% Tween-20 (PBST). The ELISA plate wells were blocked with 0.5% BSA in PBST for 1 h at 37 °C. This was followed by adding either the plasma or lung isolated immunoglobulins at a starting concentration of 100 μg/ml, 100 μl/well and serially diluting at 1:5, giving a final range from 100 to 0.8 μg/ml. Plates were incubated for 1 h at 37 °C. Finally, the plates were incubated for 1 h at 37 °C with the class-specific secondary antibodies as described for the western blotting protocol and detected using TMB substrate. The final titration curves were compared between samples as Area Under the Curve (AUC) readouts.

## Mycobacterium tuberculosis (*Mtb*) culture

Standard *Mtb* H37Rv and bioluminescent *Mtb* H37Rv[88] cultures were maintained in Middlebrook 7H9 medium (Difco) supplemented with 10% OADC, 0.2% glycerol, 0.01% Tyloxapol (SIGMA). For the bioluminescent H37Rv *Mtb* the media was supplemented with 25 μg/ml kanamycin. Cultures were grown to $1 \times 10^8$ CFU/ml *Mtb* (OD = 0.6) at which point they were suitable for infections using a multiplicity of infection (M.O.I.) of 0.1.

## Bio-electrospray (BES) 3D culture biomimetic model

A 3D culture biomimetic model, designed to mimic the host-pathogen interaction[64,89], was used to assess the impact of B cell depletion or antibody supplementation on the growth of *Mtb*.

Freshly isolated PBMCs were used for the BES experiments. Cells were infected with *Mtb* overnight in a T75 flask whereafter they were detached, washed, and mixed with 1.5% sterile alginate (Pronova UP MVG alginate, Nova Matrix) with human collagen (Advanced BioMatrix) to give a final concentration of $5 \times 10^6$ cells/ml. The PBMCs were encapsulated using an electrostatic bead generator (Nisco, Zurich, Switzerland) at a 10 ml/h flow rate using a Harvard syringe fitted with a 0.7 mm external diameter nozzle and dropping into a 100 mM CaCl$_2$ ionotropic HBSS gelling bath as described in detail by Tezera et al.[64,89]. The resulting ~600 μm diameter

microspheres containing the *Mtb* infected PBMCs were washed twice with HBSS with Ca$^{2+}$/Mg$^{2+}$ and transferred to RPMI 1640 supplemented with 10% human AB serum, 25 μg/ml kanamycin, 1% ampicillin. The beads were transferred to Eppendorffs and incubated at 37 °C, 5% CO$_2$. The use of a bioluminescent *Mtb* allowed growth to be monitored using a luminometer (GloMax 20/20 Luminometer, Promega), with readings taken every 2 to 3 days over a 15-day period.

The initial set of BES experiments involved a B cell depletion step using a EasySep™ CD19 positive selection kit II (STEMCELL), with the control PBMCs remaining "untouched". B cell depletion was performed prior to *Mtb* infection as described above. To confirm the CD19 depletion efficacy, samples were stained with a basic flow cytometry antibody panel, using L/D, CD45, CD3 and CD19 antibodies as listed in Supplementary Table 1. Cultures were maintained for a total of 15 days post infection and luminescence was measured every 2 to 3 days.

For the second set of BES experiments, PBMCs were left untouched, however, *Mtb* was treated with the antibodies or control non-specific human IgG (SIGMA, cat. 56834-25mg) for 1 h prior to infection of the PBMCs overnight and a media-only control was included. Relative infectivity was measured immediately, or cultures within microspheres were maintained for 15 days, and luminescence was measured as before.

## Seq-Well single-cell RNA sequencing

LMCs were prepared for seq-well as previously described[90]. Cells were diluted to a single cell suspension of 15,000 cells in 200 μl RPMI, 10% FBS and loaded onto a polymethylsiloxane (PDMS) array pretreated with R10 media for 15 min. Once the cells settled into the microwells, the microarray was washed with PBS (SIGMA) then sealed with a plasma functionalized polycarbonate membrane (Sterlitech). The sealed arrays were incubated at 37 °C for 40 min, followed by 20 min at room temperature in a guanidinium thiocyanate (SIGMA), EDTA (ThermoFischer), 1% β-mercaptoethanol (SIGMA) and sarkosyl (SIGMA) buffer. Capture beads were then allowed to hybridize with released mRNA in a hybridization PBS, NaCl (Thermo-Fischer), MgCl$_2$ (SIGMA) and polyethylene glycol (PEG) (SIGMA) buffer, with gentle shaking at 60 rpm for 40 min. The beads with the captured mRNA were collected from the wells with three washes of a Tris-HCl (ThermoFischer) buffer containing NaCl and MgCl$_2$ and centrifugation at $2500 \times g$ for 5 min after each wash.

cDNA synthesis was done in a mastermix containing Maxima H Minus Reverse Transcriptase, Maxima buffer, dNTPs, RNAse inhibitor, a template switch oligonucleotide and PEG for 30 min at room temperature, followed by an overnight incubation at 52 °C with end-over-end mixing. The cDNA was digested with exonuclease and complementary DNA denatured from the capture beads with a 5 min incubation in NaOH (SIGMA) and a wash with Tris-HCl, EDTA and Tween-20 (Thermo-Fischer). PCR amplification was then performed following suspension of the beads in a mastermix containing Klenow Fragment (NEB), dNTPs, PEG and dN-SMRT oligonucleotide and incubating for 45 min at 38 °C. Hereafter the PCR product was cleaned twice with an AMPure XP SPRI (Agencourt) bead cleanup at 0.6 and 0.8 times sample volume ratios. The product quality was assessed by Agilent Tape station hsD5000, confirming an expected 1 kbp amplicon with little or no primer dimers below 200 bp. The DNA libraries were quantified (Qubit High Sensitivity DNA kit) before preparation for Illumina sequencing using the Nextera XT DNA sample preparation kit, with 900 pg of each library added to the tagmentation reaction. The amplified product was cleaned again using the AMPure XP SPRI beads and the final libraries pooled before loading onto the NovaSeq 6000 using a paired-end read structure with a read 1 primer: read 1: 20 bases, read 2: 50 bases, read 1 index: 8 bases.

The sequencing reads were analysed by aligning them to the hg19 genome assembly and processing them according to the Drop-Seq Computation Protocol v2.0 (https://github.com/broadinstitute/Drop-seq). This yields a cell by gene matrix which was then transformed to log$_e$(UMI + 1) using the Seurat R package v3.1.0 (https://satijalab.org/seurat/) and scaled by a factor of 10,000. The overall quality was assessed by the read

distribution, the number of transcripts, and genes per cell. A Uniform Manifold Approximation and Projection (UMAP) was used for dimensionality reduction with the min distance set to 0.5 and the neighbours set to 30. To cluster cells with similar transcriptional profiles, an unsupervised clustering algorithm, FindClusters, was used with the resolution set to 0.5. These clusters were then further divided using a differential expression test, FindAllMarkers, in Seurat, setting "test.use" to "wilcox", Wilcoxon-adjusted $p$ value cutoff <0.01.

## Statistics and reproducibility

All analyses, except for Seqwell data, were performed in Prism (v9; GraphPad Software Inc., San Diego, CA, USA). Nonparametric tests were used throughout, with Mann–Whitney and Wilcoxon tests used for unmatched and paired samples, respectively, and a three-way ANOVA, Kruskal–Wallis comparison. $P$ values less than 0.05 were considered statistically significant and denoted by * ≤0.05; ** <0.01; *** <0.001 and **** <0.0001.

## Study approval

Human patients undergoing thoracotomy or lung resection surgery for TB or non-TB-associated pathologies gave written informed consent for a blood draw and tissue collection. The study protocol, data collection tools and associated consent forms were approved by the University of KwaZulu-Natal Biomedical Research Ethics Committee (BE 019/13). The CUBS study protocol for blood collection from healthy donors and patients with active TB was also approved (BE 022/13). Healthy blood donor study ethical approval was provided by the National Research Ethics Service Committee South Central—Southampton A, ref 13/SC/0043, Southampton, United Kingdom. Participant compensation was approved by the relevant ethics committees.

## Reporting summary

Further information on research design is available in the Nature Portfolio Reporting Summary linked to this article.

## Data availability

The source data behind the graphs in the paper can be found in Supplementary Data 1. Further inquiries can be directed to the corresponding author. The raw single-cell RNA sequencing data have not been shared publicly due to ethical restrictions, given the at-risk nature of people with HIV, but are available on request through the corresponding authors and their contact to the IRB, Prof. Shenuka Singh (Singhshen@ukzn.ac.za) or the UKZN ethics board (BREC@ukzn.ac.za).

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

## Acknowledgements
A.L. was supported by the Wellcome Trust (210662/Z/18/Z) and the BMGF (OPP1137006). R.K. and P.E. were supported by a Dorothy Temple Cross grant from the UK Medical Research Foundation (MRF-131-0002-RG-ELKI-C0850). P.E. was supported by the UK Medical Research Council (MR/P023754/1 and MR/W025728/1).

## Author contributions
R.K. conducted experimental work, data analysis and writing of the manuscript; P.O. provided initial data, assisted with data analysis and writing; D.M. assisted with data analysis and writing; L.T. conducted some biomimetic model experiments, assisted with data analysis and writing; M.A., I.M., M.C., A.N. and M.M. assisted in sample preparation; M.C. assisted with "R" analyses; F.K., K.K. and R.M. obtained samples and analysed clinical information; K.L. and K.N. assisted with microscopy; A.S. established the lung cohort; A.K.S. assisted with data analysis; P.E. helped with data interpretation, study design, and manuscript preparation; and A.L. is the senior author who designed and implemented this study, analysed the data, and cowrote the manuscript with R.K.

## Competing interests
The authors declare no competing interests.
