## [Peer Review File · Communications Biology]

Reviewers' comments:

Reviewer #1 (Remarks to the Author):

1. Brief summary of the manuscript

In this manuscript, the authors have tried to dissect the dichotomy of role of B cells in tuberculosis disease. B cell subsets identified so far have implications in diseases and physiology but remains elusive in terms of its role in protection against M.tb infection. Authors uses existing tools and information to evaluate the status B cells and its subsets of lung tissue and compare it with matched PBMCs collected from non-TB and TB-infected individuals. The study reports 7 distinct subsets with different gene signature. Using this information they phenotypically characterized these subsets establishing direct correlation of spatio-temporal location of the cells during various stages of M.tb infection. The study draws attention towards possible role of IgM as during M.tb infection.

2. Overall impression of the work

Authors have been thoughtful in designing and execution of experiments. They have taken care of minute details of B cell subset characterization at tissue sites. Head-to-head comparison of lungs with matched PBMCs in terms of B cell subsets was extensive and impressive. The use of 3D granuloma model is impressive, the study will promote its vast acceptance and will resolve animal model related ethical issues. Overall the study is a nice informative article. Also, this opens a vast avenue for B cell researchers in TB including the group itself.

3. Specific comments, with recommendations for addressing each comment

Major comments:

1. Line 106-107- Did authors perform any confirmatory staining such as IF or IHC to validate presence of DC?

2. In fig 8 F-H, author show opsonization hence enhanced phagocytosis. But there is no real-time evidence of M.tb being intracellular in their case, there could be many sticking to cell surface not to be confused with phagocytized ones. They could have performed microscopy to back their findings. In this same experiment, author uses antibiotic cocktail to culture the M.tb infected microsphere. What are the possibilities of it affecting M.tb growth. Could this be an explanation to Fig 8H?

3. In Fig 7C- Why TB patient samples are not as good as control? Please discuss the valid reason behind huge spread

Minor comments:

1. Issue with line numbers, seems like insertion of figures interrupted the sequence. Please rectify.

2. Author affiliation need attention, it seems to have mixed up (line 7-23)

3. Line 35- 'TB disease lung...' – correct

4. Line 130-131- 'lung cohort' meaning not clear

5. Revise line 212-215 for better clarity; current statement is complex

6. Line 231- "The same trend was observed for all antibody classes" does not matches the data, kindly tone down the statement.

7. Line 867- 'Pop' should be 'top'?

8. Authors must explicitly mention experimental repeats or 'n' for every figure in respective figure legend for clarity.

9. Ethical statement about whether the donors/patients were compensated should be mentioned in 'study approval' section.

10. Authors must include statement about histological analysis in the data analysis section- whether it was performed in blinded manner or not?

Reviewer #2 (Remarks to the Author):

I have read the manuscript by Krause et al with great pleasure and although the data are interesting and contribute novel insights into our understanding of B-cell responses locally in the lung, I do have several questions that need to be resolved.

- The description of patients is insufficient, details need to be added on numbers of patients included into the study, overlap between patients for the different analyses. Demographic data, eg age, gender etc are lacking. Related the description of the control population is insufficient, it is mentioned that these are lung cancer patients, but is anything known on their IGRA status? Is LTBI excluded?
 - How is the lung tissue selected? In lung surgery often lobes are removed, are entire lobes homogenized for cell isolation or are regions with macroscopic lesions selected? This will be in particular relevant for the TB patients, how much of the immune cells are estimated to be of granuloma origin?
 - For scRNAseq 9 TB and 3 non TB patients were selected, how were these selected? The data seem to describe the TB patients only, were any comparisons to the non-TB group made? The expression of Ig genes is relatively low in most subsets, was that expected?
 - Details should be added to figure legends, eg in Figure 2 5 patients are analysed, was that a single granuloma of each patient? How were the sizes for quantification determined, GrALT shapes seem irregular, how were edges determined to calculate percentage of DAB staining? Adding up percentages seems to reach more than 100% whereas not all cells are stained, how is this explained?
 - Which LNs were sampled for the data collection in Figure 3D? are these lung draining LN?
 - Figure 5 would benefit from some more detailed explanation, are these pooled data from all donors? Is this blood or lung? Which number of donors was used here? Are they the same as in figure 1?
 - Figure 6, are these TB patients only or also controls?
 - In Figure 7, the TB patients show variable effects of B cell depletion, however all data are related to the undepleted samples. Did the authors observe differences between patients and controls in the undepleted samples? Could this explain part of the observed effect? Eg. Do unfunctional B cells in TB patients hamper uptake/elimination in the undepleted sample? If so, no differences may be observed upon depletion.
 - What is the source of the non-specific Ig in Figure 8?
 - In isolation of the lung Ig was there any correction for the lung volume started with? Was the total yield of Ig comparable between groups of patients? Could responses to other antigens, eg tetanus or flu be detected in all groups of individuals?
 - Do the authors think there is any relation to the inflammatory status of the TB patients? Eg would some data be similar if patients with other inflammatory disease would be analysed? This should be discussed.
 - As questioned above, it seems that the sampling is rather crude, i.e. by homogenizing whole lung parts, how would that affect data interpretation? Could that potentially induce sampling errors and thereby underrepresentation of the antigen specific B cell responses?
- The lung sampling although described as lung mononuclear cell isolation seems not to contain any purification of leucocytes, what is the proportion of epithelial cells that are in suspension as well? Where any markers eg cell activation assessed on the epithelial cells?

Reviewer #3 (Remarks to the Author):

This study examines B cell phenotype and function within the lung tissue of TB patients undergoing lung resection surgery, compared to nonTB subjects undergoing similar surgery from other causes. This is a very complex analysis of B cell subsets in the lung tissue and is clearly a huge amount of work. My only caution for the authors is that the data is largely descriptive, with cell subsets being suggested rather than being proved to be those cell subsets. I would prefer a more unbiased approach

to describing these subsets, using terms such as GC-like or ASC-like. This would be the major critique I have of this otherwise comprehensive analysis of human B cell subsets within TB-diseased lung tissue.

Minor issues

Line 86: "where" should be "were".

Figure 4: It is not clear what the benefit of this analysis is, other than to perhaps get an overview of differentially expressed genes in B cells that might indicate their role in the tissue. But there is no correlation of this Figure with that of Figure 5 which begs the question: which analysis is (more) correct? Perhaps some discussion on this point is warranted.

In addition, the authors mention several genes that are upregulated in this analysis, and that are also present on B cells in GCs. But since this data is from lung tissue and NOT from a GC, it is unclear what the relevance of GC markers are for B cells in lung tissue. Moreover, CD10 is used as a marker of GC, but is also a more traditional marker for transitional B cells. These cells could just as easily be transitional B cells in lung tissue as those derived from a GC and appearing in the lung. Furthermore, CD10, although expressed, is not highly expressed compared to other markers. Perhaps a more cautious approach as to specific B cell subsets would be better here. Terms such as GC-like B cells or some such would appease the more sceptical B cell immunologists out there.

Figure 5: What are the DN markers? The authors mention CD38/IgD/IgM as negative markers, but they do not define which two of these three are used to define the DN term. This should be rectified.

Line 277: This sentence is misleading and should be modified. Th1 and Th17 cells have only been identified as such in one study, in macaques, and using IV BCG administration. To suggest this is THE correlate of protection in humans is a gross exaggeration of the data. In any case, the authors do not even need to lead with this erroneous text and could simply just delete it. If not, the authors should modify the text to better indicate the ambiguity and unknown "correlate of protection" in humans.

Lastly, this likely reflects personal preference, but I do find the Discussion to be a little exaggerated in over-estimating the role of B cells in TB disease, and indeed in making some tenuous connections with other diseases. A case in point: from Line 294 "In addition, 294 atypical or ABC are associated with many autoimmune conditions including Lupus and multiple sclerosis, and TB disease shares clinical similarities with such autoimmune conditions. It is likely, therefore, that the atypical B cells enriched in TB diseased lungs play an important role..." TB is a bad enough disease without invoking tenuous links to autoimmune diseases, and we really do not know what atypical B cells do, in any animal model, because they have been so understudied. I feel the Discussion would benefit from toning down the "importance" of the B cell subsets that have been putatively described and not definitively identified. But this is perhaps more stylistic than anything else, so up to the authors as to whether they would like to rewrite the Discussion to reflect the more probable and nuanced role of B cells in the lung and TB disease.

Reviewers' comments:

Reviewer #1 (Remarks to the Author):

1. Brief summary of the manuscript

In this manuscript, the authors have tried to dissect the dichotomy of role of B cells in tuberculosis disease. B cell subsets identified so far have implications in diseases and physiology but remains elusive in terms of its role in protection against M.tb infection. Authors uses existing tools and information to evaluate the status B cells and its subsets of lung tissue and compare it with matched PBMCs collected from non-TB and TB-infected individuals. The study reports 7 distinct subsets with different gene signature. Using this information they phenotypically characterized these subsets establishing direct correlation of spatio-temporal location of the cells during various stages of M.tb infection. The study draws attention towards possible role of IgM as during M.tb infection.

2. Overall impression of the work

Authors have been thoughtful in designing and execution of experiments. They have taken care of minute details of B cell subset characterization at tissue sites. Head-to-head comparison of lungs with matched PBMCs in terms of B cell subsets was extensive and impressive. The use of 3D granuloma model is impressive, the study will promote its vast acceptance and will resolve animal model related ethical issues. Overall the study is a nice informative article. Also, this opens a vast avenue for B cell researchers in TB including the group itself.

3. Specific comments, with recommendations for addressing each comment

We thank the reviewer for their overall positive evaluation of our work.

Major comments:

1. Line 106-107- Did authors perform any confirmatory staining such as IF or IHC to validate presence of DC?

We thank the reviewer for this comment. CD21 was used as a marker for Dendritic Cells as has been done in other published studies (Slight *et al.*, 2013; Gars *et al.*, 2020). However, as mentioned in the text that CD21 is also expressed by mature B cells as described by Slight *et al.* (1998), and therefore it is possible that CD21 staining observed is not limited to DCs.

We did not show this originally, but when zoomed in, the CD21 staining has a distinct “dendrite-like” pattern, whereas the CD20 and CD3 stains are more lymphoid in shape. In addition, CD21 cells are contained in a specific region of the follicle and not present throughout the entire follicle. We believe these factors are suggestive of the presence of DCs but we can't rule out other CD21 expressing cell types at this time. Additional staining is on-going but is not currently available. To address the comment, we now include a supplementary figure showing the zoomed in staining pattern of CD21.

We have included the following amendments in the manuscript:

An additional supplementary figure on page 41 (**Supplementary Figure 1**) to highlight the differential staining patterns of anti-CD20, anti-CD3 and anti-CD21.

The following description of the results was also included in the text (lines 109-113): “Of interest were the staining patterns of anti-CD20, anti-CD3 and anti-CD21 in particular (**Supplementary Figure 1**). CD20 and CD3 staining are characteristic of lymphoid cells whereas CD21 stained cells displaying a more dendrite-like pattern. The CD20⁺ cells occupy most of the aggregate, whereas the CD3⁺ cells are more peripheral, and the CD21⁺ cells seem to occupy the centre of the aggregate.”

Supplementary Figure 1: Immunohistochemistry of human lung tissue resection from a TB patient demonstrates cellular organisation within aggregates associated with the granuloma. Serial sections of TB patient lung tissue were stained for B cells (anti-CD20), T cells (anti-CD3) and dendritic cells/mature B cells (anti-CD21). The boxed region is enlarged to illustrate the distinct staining patterns of the different markers within the granuloma associated aggregates.

2. In fig 8 F-H, author show opsonization hence enhanced phagocytosis. But there is no real-time evidence of M.tb being intracellular in their case, there could be many sticking to cell surface not to be confused with phagocytized ones. They could have performed microscopy to back their findings.

We thank the reviewer for making this important point. Unfortunately, we did not perform confocal microscopy to confirm that opsonised bacteria are phagocytosed. However, after infection, the cells are extensively washed, which from previous work has proved effective at removing extracellular bacteria. In addition, we have previously shown that extracellular bacteria in the 3D model grow very slowly (Tezera et al., *elife*, 2017). We therefore think it is highly unlikely that our findings are influenced by the presence of extracellular bacteria.

We have added this clarification to the text (lines 264-267): “Although the flow cytometry does not determine if the bacteria are phagocytosed or cell-associated, the cells are extensively washed post-infection, and we have previously shown that *Mtb* growth in this system is predominantly intracellular, suggesting *Mtb* is primarily intracellular⁶⁴.”

In this same experiment, author uses antibiotic cocktail to culture the M.tb infected microsphere. What are the possibilities of it affecting M.tb growth. Could this be an explanation to Fig 8H?

The Luciferase plasmid required to measure *Mtb* growth by luminescence in this system contains an antibiotic-resistance cassette. The antibiotic in the media therefore ensures the luciferase gene is expressed and in previous studies did not affect *Mtb* growth (<http://www.ncbi.nlm.nih.gov/pubmed/20520722>). This media is the same in all experimental conditions and therefore would be highly unlikely to drive the differences observed here.

3. In Fig 7C- Why TB patient samples are not as good as control? Please discuss the valid reason behind huge spread

We believe these data are consistent with disruption of the B cell compartment during TB disease, but our data do not give a clear idea of the potential mechanism. Nonetheless these findings are consistent with published literature, which we attempted to highlight in the discussion as follows (lines 345-350):

“This suggests that B cells can contribute to control of *Mtb* growth, but also that the B cell compartment is disrupted during active TB. In a longitudinal study, Moreira-Teixeira *et al.* observed a decreased B cell signature with increased disease progression in the blood of LTBI patients, LTBI-progressors and ATBI patients, consistent with our observation¹⁶. Similarly, several studies showed that the proportions of B cell populations and transcriptional signatures are affected by TB^{11,15,16,79}.”

However, we did not speculate on why the data generated from TB patients was so variable. In addition, reviewer 2 asked a question about *Mtb* growth in non-depleted PBMC from TB patients vs controls. Interestingly, these data showed elevated *Mtb* growth in PBMC from TB-infected patients. There are several potential mechanisms driving this, including the potential effect of dysregulated B cells. We have now added this to the discussion section (lines 339-345) and include data on *Mtb* growth in the absence of B cell depletion as Figure 7A.

“We believe it is likely to reflect the fact that the TB patients were, by chance, at very different stages of disease severity. For example, in more severe TB disease we expect higher frequencies of pro-inflammatory atypical B cells, which may contribute to *Mtb* growth. The depletion of B cells in these cases may reduce *Mtb* growth, both in the 3D model and by analogy, in the lung. This is supported by the fact that *Mtb* growth was higher in PBMCs from TB patients vs controls before B cell depletion (**Fig. 7A**). However, additional experiments are required to explore this hypothesis further.”

Minor comments:

1. Issue with line numbers, seems like insertion of figures interrupted the sequence. Please rectify.

This seems to be a technical issue with the insertion of the figures and possibly with the conversion of the manuscript from Microsoft Word to a PDF document. The digital version line numbers do not shift.

2. Author affiliation need attention, it seems to have mixed up (line 7-23)

This has been addressed.

3. Line 35- 'TB disease lung...' – correct

This was corrected and now reads "TB diseased lung..." as opposed to "TB disease lung tissue...".

4. Line 130-131 (now line 139)- 'lung cohort' meaning not clear

This statement was rewritten as follows: "analysis of additional lung tissue samples obtained from the same lung cohort." The cohort referred to is described in the methods section and summarised in the cohort Table 1. Please refer to reviewer 2 comment below.

5. Revise line 212-215 for better clarity; current statement is complex

The model uses peripheral blood mononuclear cells (PBMC) infected with bioluminescent *Mtb*, which are encapsulated in collagen/alginate microspheres, each representing a 3D granulomatous microenvironment with an extracellular matrix scaffold.

This was revised as follows (lines 229-232): "The model uses peripheral blood mononuclear cells (PBMC) infected with bioluminescent *Mtb*. These *Mtb*-infected PBMC are then encapsulated in collagen/alginate microspheres that contain an extracellular matrix scaffold. Each microsphere therefore represents a 3D microenvironment that mimics an individual granuloma."

6. Line 231- "The same trend was observed for all antibody classes" does not match the data, kindly tone down the statement.

The same trend was observed for all antibody classes, reaching significance for IgM (**Fig. 8E**).

This was revised as follows (lines 254-255): "Although individual *Mtb*-reactive antibody classes showed similar upregulation, only IgM was significantly upregulated (**Fig. 8E**)."

7. Line 867- 'Pop' should be 'top'?

This was an abbreviation for the "populations" but has been removed for clarity.

8. Authors must explicitly mention experimental repeats or 'n' for every figure in respective figure legend for clarity.

This was added to each respective figure.

9. Ethical statement about whether the donors/patients were compensated should be mentioned in 'study approval' section.

"Participant compensation was approved by the relevant ethics committees." was added to the study approval section as requested (lines 601-602).

10. Authors must include statement about histological analysis in the data analysis section- whether it was performed in blinded manner or not?

The text was amended as follows (line 434): “DAB stained cells were quantified using QuPath (<https://gupath.github.io/>) with the operator blinded to the nature of the sample.”

Reviewer #2 (Remarks to the Author):

I have read the manuscript by Krause et al with great pleasure and although the data are interesting and contribute novel insights into our understanding of B-cell responses locally in the lung, I do have several questions that need to be resolved.

- The description of patients is insufficient, details need to be added on numbers of patients included into the study, overlap between patients for the different analyses. Demographic data, eg age, gender etc are lacking. Related the description of the control population is insufficient, it is mentioned that these are lung cancer patients, but is anything known on their IGRA status? Is LTBI excluded?

We thank the reviewer for these suggestions and have added a descriptive table of the cohort as Table 1 in the methods section on page 17:

Table 1: Descriptive table of the study cohort (n=76):

	Gender	+ Lymph node	HIV +ve	TB	Smokers	Age*
Male	47 (62%)	7	20 (43%)	34 (72%)	21 (45%)	42 (35-48)
Female	29 (38%)	10	19 (66%)	24 (83%)	1 (3%)	39 (34,47)
Total	76	17	39 (51%)	58 (76%)	22 (29%)	

* Median and IQR are indicated

The following text was included in the methods section (lines 400-406):

“The cohort consisted largely of a Black African population 82% (90% of males and 74% of females). A total of 76 lung samples were collected, including 17 lung draining lymph nodes and all samples with matched blood. About half of the cohort was HIV positive, with positivity skewing slightly toward females. The majority of resections were TB-related (76%), with a median age of 41 years, and a relatively similar age distribution amongst males and females. Close to two thirds of cases were males, and of these 45% were active/previous smokers. Lung cancer resections were dissected, and the healthy tissue margins were retained as the TB-negative control lung samples.”

The Results section was amended to describe the control samples were not IGRA tested (lines 92-94): “These controls were not IGRA tested and thus latent TB infection could not be excluded, although TB prevalence is so high in KwaZulu-Natal, most donors are highly likely to have been TB-exposed.”

- How is the lung tissue selected? In lung surgery often lobes are removed, are entire lobes homogenized for cell isolation or are regions with macroscopic lesions selected? This will be in particular relevant for the TB patients, how much of the immune cells are estimated to be of granuloma origin?

We refer to the sample collection in the methods section (lines 390-392): “Tissues were collected following surgery to remove the irreversibly damaged lobes or lungs. Resected lung tissue was dissected into smaller

tissue samples on site and transferred to RPMI 1640 media containing 10% FBS and kept on ice. Samples were received for processing within one hour of dissection and processed immediately in a BSL3 facility.”

The following was added to clarify the sample selection (lines 392-395): “The sections selected for analysis represented visibly low-, mid- and severely diseased areas of the lung, which were combined and processed. Therefore, the isolated cells were representative of the entire lung and were not necessarily only of granulomatous origin.”

- For scRNAseq 9 TB and 3 non TB patients were selected, how were these selected? The data seem to describe the TB patients only, were any comparisons to the non-TB group made? The expression of Ig genes is relatively low in most subsets, was that expected?

We thank the reviewer for picking up this confusing presentation. On reflection, the non-TB patient samples contributed only 5 B cells of the 409 total B cells in the analysis. Consequently these unhelpful data have now been omitted and only the TB patient-derived cells are now shown. We have clarified this in the text (lines 139-141): “Of a total 20962 single cells recovered from 9 resected TB patient lungs, 404 displayed canonical B cell markers including the lineage markers CD19 and CD20.”

The immunoglobulin expression levels were low in the naïve and germinal centre subsets, however increased in the phenotypes that resembled maturing plasma blasts and plasma cells especially. This was expected as in the early stages of development the B cells would express immunoglobulin as their B cell receptor only on their surface, whereas in the case of plasma blasts and plasma cells the immunoglobulins are continuously expressed and secreted from the cells as soluble immunoglobulins. The following statement is now added (lines 158-160):

“The immunoglobulin and CD79a transcript abundance generally increased from populations 0 to 6, and since both are associated with the B cell surface receptor, this also suggests the local B cell population is maturing^{9,50}”

- Details should be added to figure legends, eg in Figure 2 5 patients are analysed, was that a single granuloma of each patient? How were the sizes for quantification determined, GrALT shapes seem irregular, how were edges determined to calculate percentage of DAB staining? Adding up percentages seems to reach more than 100% whereas not all cells are stained, how is this explained?

The total slide areas were enumerated, therefore the legend was amended as follows (lines 862-865): “(B) The diaminobenzidine (DAB) stained cells were enumerated and expressed as % DAB positive cells of total nucleated (haematoxylin+) cells, per total slide area imaged. The relative frequencies from a total of five different TB patient tissue samples are compared.”

The entire slide was quantified per patient to avoid selection bias for particular regions. The different cell surface markers are co-expressed by certain cells and therefore cannot be added up to give a 100%. Only the hemozoin fraction of cells would add up to 100%.

We have amended the text in line 116 to state “As some surface markers are co-expressed by different cells, the total exceeds 100%”.

- Which LNs were sampled for the data collection in Figure 3D? are these lung draining LN?

We apologise for not describing this clearly. All data derived from lung draining lymph nodes. This was mentioned in the results section, however, to clarify this further we have amended the following in text:

This was corrected in Figure 3 legend (line 892), where “derived” was replaced with “draining”: “(D) This analysis was extended to lung draining lymph nodes (LN, n=15) from the TB cohort for comparison.”

The following comment was included in the methods section to clarify this (lines 453-454): “Lung draining lymph nodes were processed similarly without the need for GentleMACs homogenization or collagenase D and DNase I treatment.”

- Figure 5 would benefit from some more detailed explanation, are these pooled data from all donors? Is this blood or lung? Which number of donors was used here? Are they the same as in figure 1?

It is mentioned in the legend that this is a representation from a single patient. This is a combination of blood and lung and is now explained in the figure legend as follows (lines 951-956):

“The different B cell populations were clustered spatially using tSNE, with a single patient presented here, followed by a key describing the major phenotypes each of which is assigned a different colour. The relative expression of the specific markers associated with each phenotype were plotted alongside in a heatmap. Since the blood and lung samples were concatenated into a single file, the relative frequencies of cells in the blood or lung tissue compartment could be compared as shown alongside the heatmap.”

These are not the same samples shown in Fig.1 and this was further clarified in the results section (lines 173-176): “We obtained 13 additional lung samples and analysed lung homogenate and matched blood samples by flow cytometry using three separate antibody panels, each elaborating on a core set of shared markers included in **Fig. 1 and 3**: CD19, CD27, CD38, IgD and IgM (**Fig. 5**).”

- Figure 6, are these TB patients only or also controls?

These are TB patients only, as stated in the figure legend and in the results section. To clarify this further we changed the figure title as follows (lines 947-948): **Figure 6: tSNE-derived unique B cell phenotypes associated with the blood or lung compartment of TB patients.**

- In Figure 7, the TB patients show variable effects of B cell depletion, however all data are related to the undepleted samples. Did the authors observe differences between patients and controls in the undepleted samples? Could this explain part of the observed effect? Eg. Do unfunctional B cells in TB patients hamper uptake/elimination in the undepleted sample? If so, no differences may be observed upon depletion.

We thank the reviewer for this very helpful suggestion which we had not considered. Overall the growth rate in the TB+ patient PBMCs is greater than that in the TB- patient-derived PBMCs when B cells are not depleted. This has now been included as a **Figure 7A** and descriptive text is included in the results section as follows (lines 235-237):

“Without B cell depletion, the overall *Mtb* growth rate was significantly greater in the TB patient derived undepleted PBMC versus the control samples, consistent with immune dysregulation during TB disease (**Fig. 7A**).”

“Figure 7: B cells contribute to host control of *Mtb* in healthy donors in a 3D biomimetic model, but in TB patients the effect is highly variable. (A) Comparatively, *Mtb* grew more rapidly in TB patient derived PBMC. B cells were depleted from PBMCs, of which four donors were healthy and four had active TB, and compared to undepleted PBMCs. (B) Representative FACS plots of the depletion process. (C) B cell depletion leads to significantly higher *Mtb* growth in a healthy donor PBMC sample, representative of four healthy donors. (D)

Mtb outgrowth data at day 7 from a total of four PBMC donors from each group are summarized, demonstrating the highly variable effect of B cell depletion in TB patients. Data were normalized to the undepleted control group and compared by Wilcoxon matched-pairs signed rank test. *P* values are denoted by * ≤ 0.05 ; ** < 0.01 and **** < 0.0001 .”

In addition, these data are now mentioned in the discussion section (lines 339-345) with relation to potential mechanisms behind the variable effect of B-cell depletion in TB diseased individuals.

“We believe it is likely to reflect the fact that the TB patients were, by chance, at very different stages of disease severity. For example, in more severe TB disease we expect higher frequencies of pro-inflammatory atypical B cells, which may contribute to *Mtb* growth. The depletion of B cells in these cases may reduce *Mtb* growth, both in the 3D model and by analogy, in the lung. This is supported by the fact that *Mtb* growth was higher in PBMCs from TB patient’s vs controls before B cell depletion (Fig. 7A). However, additional experiments are required to explore this hypothesis further.”

- What is the source of the non-specific Ig in Figure 8?

We apologise for omitting the important information. The details of the source of non-specific immunoglobulin were included in the methods section as follows (line 544-546): “For the second set of BES experiments, PBMCs were left untouched, however, *Mtb* was treated with the antibodies or control non-specific human IgG (SIGMA, cat. 56834-25mg) for one hour prior to infection of the PBMCs overnight.”

- In isolation of the lung Ig was there any correction for the lung volume started with? Was the total yield of Ig comparable between groups of patients? Could responses to other antigens, eg tetanus or flu be detected in all groups of individuals?

Since the size of lung sections varied between patients, the total yields of antibodies isolated did vary. Therefore, to correct for this an equal concentration of antibodies was used for the ELISA assays across all patients. This was set at 100 $\mu\text{g/ml}$ and serially diluted 1:5 to give a range from 100 to 0.8 $\mu\text{g/ml}$ as stated in the methods section.

Figure 8 legend was amended to clarify this (lines 1035-1036): “An equal concentration range (100 to 0.8 $\mu\text{g/ml}$) of isolated antibody was used to standardize the assay and allow for comparison between patients.”

We thank the reviewer for the idea of testing the antibody reactivity to alternate pathogens. This will form part of future work to further characterize these antibodies and compare differences between antibodies in circulation and those isolated from the lung.

- Do the authors think there is any relation to the inflammatory status of the TB patients? Eg would some data be similar if patients with other inflammatory disease would be analysed? This should be discussed.

We thank the reviewer for this point. We do discuss the similarities to some autoimmune conditions such as Lupus and multiple sclerosis (lines 325-327). This was also raised by reviewer 3’s final comment and we therefore address these points together.

- As questioned above, it seems that the sampling is rather crude, i.e. by homogenizing whole lung parts, how would that affect data interpretation? Could that potentially induce sampling errors and thereby underrepresentation of the antigen specific B cell responses?

Due to the valuable nature of the samples, none of the tissue was discarded and the idea was to attempt to use methods to standardise the samples for comparison. For example, with regard to the antibody based experiments all samples were used at the same concentration. Comparisons of total antibody yields would not have been possible unless the initial samples were standardized by weight or size or volume for example. This was to correct for any sampling variance. Concerning the B cell phenotyping and comparison of population frequencies, these are all related to the number of CD45 positive cells detected and this therefore again allows

for comparison between samples. Although the total number of CD45 cells may vary, comparing the fraction of cells positive for a specific marker relative to a parent population makes these comparable.

In combination with data from entire lung, we also perform spatial-specific analysis using IHC, and normalisation to total parent populations such as CD45, and so the analysis of overall lung, spatial localisation and immune phenotypes gives a normalised assessment of B cell functions within the constraints of working with human surgical tissue which is inherently variable between donors.

The lung sampling although described as lung mononuclear cell isolation seems not to contain any purification of leucocytes, what is the proportion of epithelial cells that are in suspension as well? Were any markers eg cell activation assessed on the epithelial cells?

Unfortunately, we were not able to reliably purify leucocytes using density centrifugation or magnetic beads. Therefore, the anti-CD45 (common leukocyte antigen) was included in the flow cytometry panels to allow us to report immune cell frequency as a fraction of total leukocytes. No flow cytometry markers were included for non-hematopoietic cells in this study. However, single cell sequencing of lung homogenate has shown the presence of such cells, including fibroblasts, endothelial and other stromal subsets as expected. This manuscript is currently in preparation.

We included the following in text (lines 141-143): “We focused on leukocytes, although scRNAseq analysis demonstrated the presence of other stromal cells in the lung homogenates such as epithelial cells and fibroblasts (manuscript in preparation).”

Reviewer #3 (Remarks to the Author):

This study examines B cell phenotype and function within the lung tissue of TB patients undergoing lung resection surgery, compared to nonTB subjects undergoing similar surgery from other causes. This is a very complex analysis of B cell subsets in the lung tissue and is clearly a huge amount of work. My only caution for the authors is that the data is largely descriptive, with cell subsets being suggested rather than being proved to be those cell subsets. I would prefer a more unbiased approach to describing these subsets, using terms such as GC-like or ASC-like. This would be the major critique I have of this otherwise comprehensive analysis of human B cell subsets within TB-diseased lung tissue.

We thank the reviewer of their overall positive assessment and accept that the analysis of this human lung tissue is by nature primarily descriptive, and so have amended specific wording as the reviewer suggests.

Minor issues

Line 86: “where” should be “were”.

This was corrected.

Figure 4: It is not clear what the benefit of this analysis is, other than to perhaps get an overview of differentially expressed genes in B cells that might indicate their role in the tissue. But there is no correlation of this Figure with that of Figure 5 which begs the question: which analysis is (more) correct? Perhaps some discussion on this point is warranted.

We agree with the reviewer’s point that it would have been ideal to directly correlate scRNAseq data with FACS analysis to confirm the presence of putative B cell populations. Unfortunately, this was not possible due to the challenges of carrying out this study using fresh human surgical samples as concurrent analysis is technically impossible. However, as there is so little data available from TB diseased human lungs we believe these data are of interest. Specifically, although not directly correlated, the fact that we see similar B cell populations at both the transcriptional and protein level (FACS) is highly supportive and, overall, is consistent

with the presence of functioning B cell follicles within TB-diseased lungs. For these reasons, we believe these data are valuable together.

We now comment on lines 285-288: "For technical reasons, it was not possible to perform scRNAseq analysis and FACS analysis on the same samples, the fact that we observe similar changes at both the transcriptional and protein level supports the phenotypic changes observed by each approach, despite analysis being in different samples".

In addition, the authors mention several genes that are upregulated in this analysis, and that are also present on B cells in GCs. But since this data is from lung tissue and NOT from a GC, it is unclear what the relevance of GC markers are for B cells in lung tissue. Moreover, CD10 is used as a marker of GC, but is also a more traditional marker for transitional B cells. These cells could just as easily be transitional B cells in lung tissue as those derived from a GC and appearing in the lung. Furthermore, CD10, although expressed, is not highly expressed compared to other markers. Perhaps a more cautious approach as to specific B cell subsets would be better here. Terms such as GC-like B cells or some such would appease the more sceptical B cell immunologists out there.

We agree with the reviewer that CD10 is a marker on transitional B cells, however these cells would not express the markers associated with maturation, including AICDA, Bcl6 and CD79a for example. However, the reviewer's point in general about caution when describing these cell types is very well taken. We have therefore addressed the language used especially when describing the populations identified from the RNA sequencing data in the text as follows:

(Lines 167-170): "Thus, scRNAseq data supports the presence of GC-like B cell populations, ASC-like and atypical/innate-like B cells in TB diseased lung tissue. The next aim was to identify similarities between these B cell populations putatively identified by sequencing, with those identified by flow cytometry."

(Lines 223-226): "Together these flow cytometry data confirm the presence of diverse B cell phenotypes in TB diseased lung tissue, including an expanded CD11c expressing atypical B cell population, subsets potentially associated with GC activity, antibody production, and regulation."

Figure 5: What are the DN markers? The authors mention CD38/IgD/IgM as negative markers, but they do not define which two of these three are used to define the DN term. This should be rectified.

We thank the reviewer for picking this up and apologize for not being clear. The double negative (DN) B cell phenotype is defined as CD27⁻ and IgD⁻. The results section was amended to define this more clearly as follows (lines 185-188):

"Common to all three panels was a prominent B cell phenotype that was CD27^{mid/lo}, CD38⁻, IgD⁻ and IgM⁻ (**Fig. 5A green, B blue, C tan**) designated as an atypical/double negative (DN) memory population. The double negative phenotype is defined as being negative for CD27 and IgD⁵⁰. This was expanded in the lung compared to matched blood."

Line 277: This sentence is misleading and should be modified. Th1 and Th17 cells have only been identified as such in one study, in macaques, and using IV BCG administration. To suggest this is THE correlate of protection in humans is a gross exaggeration of the data. In any case, the authors do not even need to lead with this erroneous text and could simply just delete it. If not, the authors should modify the text to better indicate the ambiguity and unknown "correlate of protection" in humans.

We accept the reviewer's point and have removed this line from the text.

Lastly, this likely reflects personal preference, but I do find the Discussion to be a little exaggerated in over-estimating the role of B cells in TB disease, and indeed in making some tenuous connections with other diseases. A case in point: from Line 294 "In addition, 294 atypical or ABC are associated with many autoimmune conditions including Lupus and multiple sclerosis, and TB disease shares clinical similarities with

such autoimmune conditions. It is likely, therefore, that the atypical B cells enriched in TB diseased lungs play an important role..." TB is a bad enough disease without invoking tenuous links to autoimmune diseases, and we really do not know what atypical B cells do, in any animal model, because they have been so understudied. I feel the Discussion would benefit from toning down the "importance" of the B cell subsets that have been putatively described and not definitively identified. But this is perhaps more stylistic than anything else, so up to the authors as to whether they would like to rewrite the Discussion to reflect the more probable and nuanced role of B cells in the lung and TB disease.

We thank the reviewer for these comments and agree that toning down the language is appropriate. The discussion has been modified as follows (lines 327-332):

"It is possible, therefore, that the atypical B cells enriched in TB diseased lungs may contribute to immunopathogenesis of TB, through mechanisms including antibody and cytokine production and cross talk with T cells. Overall, the heterogeneity of B cells observed in TB diseased lung tissue is consistent with multiple putative roles in TB protection and pathogenesis. However, these data are only descriptive of the array of B cells found in the human TB lung, and specific functional roles remain to be mechanistically proven."

REVIEWERS' COMMENTS:

Reviewer #1 (Remarks to the Author):

Authors have satisfactorily addressed all queries raised. It's understood that human TB research has numerous limitations and hence authors have put together an informative manuscript. This will surely boost 'B-cell in TB' research.

Reviewer #2 (Remarks to the Author):

Thanks for the elaborate comments on my questions. I have no further questions.

Reviewer #3 (Remarks to the Author):

The authors have very reasonably considered my comments and adjusted the manuscript where necessary. They have addressed all of my concerns and I have no further issue with the manuscript. In addition, in reading the responses to other reviewers, I find that the authors have also addressed these comments adequately, including additional Tables and Figures.